# Pseudo-proxy evaluation of Climate Field Reconstruction methods of North Atlantic climate based on an annually resolved marine proxy network

Maria Pyrina[1], Sebastian Wagner[1], Eduardo Zorita[1]

[1]Helmholtz Zentrum Geesthacht, Institute of Coastal Research, Geesthacht, 21502, Germany

*Correspondence to*: Maria Pyrina (maria.pyrina@hzg.de)

**Abstract.** Two statistical methods are tested to reconstruct the inter-annual variations of past sea surface temperatures (SSTs) of the North Atlantic (NA) Ocean over the past millennium, based on annually resolved and absolutely dated marine proxy records of the bivalve mollusk *Arctica islandica*. The methods are tested in a pseudo-proxy experiment (PPE) set-up using state-of-the-art climate models (CMIP5 Earth System Models) and reanalysis data from the COBE2 SST data set. The methods were applied in the virtual reality provided by global climate simulations and reanalysis data to reconstruct the past NA SSTs, using pseudo-proxy records that mimic the statistical characteristics and network of *Arctica islandica*. The multivariate linear regression methods evaluated here are Principal Component Regression and Canonical Correlation Analysis. Differences in the skill of the Climate Field Reconstruction (CFR) are assessed according to different calibration periods and different proxy locations within the NA basin. The choice of the climate model used as surrogate reality in the PPE has a more profound effect on the CFR skill than the calibration period and the statistical reconstruction method. The differences between the two methods are clearer for the MPI-ESM model, due to its higher spatial resolution in the NA basin. The pseudo-proxy results of the CCSM4 model are closer to the pseudo-proxy results based on the reanalysis data set COBE2. Conducting PPEs using noise contaminated pseudo-proxies instead of noise free pseudo-proxies is important for the evaluation of the methods, as more spatial differences in the reconstruction skill are revealed. Both methods are appropriate for the reconstruction of the temporal evolution of the NA SSTs, even though they lead to a great loss of variance away from the proxy sites. Under reasonable assumptions about the characteristics of the non-climate noise in the proxy records, our results show that the marine network of *Arctica islandica* can be used to skillfully reconstruct the spatial patterns of SSTs at the eastern NA basin.

## 1 Introduction

Several studies have targeted to reconstruct hemispheric or global average temperature from networks of proxy records (Hegerl et al., 2007; Mann and Jones, 2003; Mann et al., 2008; Marcott et al., 2013; Moberg et al., 2005), as well as large scale temperature patterns of past changes at global (Mann et al., 2009; Rutherford et al., 2005; Wahl and Ammann, 2007) and regional scale during the last millennium (Ahmed et al., 2013; Büntgen et al., 2017; Esper et al., 2012; Luterbacher et al., 2004; Luterbacher et al., 2016; Xoplaki et al., 2005). Most of these studies have primarily used terrestrial proxy records including very few marine proxies that could contain information about the vast ocean areas, e.g. over the North Pacific and the North Atlantic Ocean (NA). In the context of marine proxy networks, reconstructions of global SSTs for the past 2000 years derived from individual marine reconstructions were recently published as a global synthesis of SST for the Common Era (CE), (McGregor et al., 2015). Due to the diversity of marine proxies used, McGregor et al., (2015) averaged each SST reconstruction into 200-yr bins providing a synthesis of proxy data representative for global variations, yet losing temporal information on decadal to multi-decadal time scales. Other studies have targeted reconstructions of spatial patterns of SST changes using marine proxies, but

mostly regarding regions in the tropics and subtropics (Dowsett and Robinson, 2009; Evans et al., 2000; Tierney et al., 2015; Wilson et al., 2006). Regarding the NA basin, Marshal et al., (2002) used sediment cores to locally reconstruct upper ocean temperature during the Holocene (10-0 ka B.P.), while Gray et al., 2004 reconstructed the Atlantic Multi-decadal Oscillation (AMO) since 1567 AD using tree ring records. Terrestrial proxy records were also used by Wang et al., (2017) for the reconstruction of a 1200-year AMO index.

In the present study, we test the potential of using a high-resolution marine proxy archive collected in the NA Ocean to reconstruct spatially resolved SST fields for the last millennium, and further test whether CFR methods are appropriate for that specific marine proxy network. The real world proxy archive that motivates our study is *Arctica islandica*. It is an absolutely dated (Scourse et al., 2006), annually resolved (Butler et al., 2009), and long lived bivalve mollusk (Butler et al., 2013; Jr et al., 2008) that can serve as a proxy for the Atlantic marine climate (Reynolds et al., 2017; Wanamaker Jr et al., 2012). It can be found in various regions in the NA, including locations north of 60° N (Dahlgren et al., 2000), and amongst others, it can be used to reconstruct SSTs (Eagle et al., 2013) and major NA climate modes like the AMO (Mette et al., 2015). *Arctica islandica* proxy collection sites have been found to contain a NA SST basin signal (Pyrina et al., 2017) which renders this proxy suitable for CFRs.

Numerous reconstructions of large scale mean climate have employed CFR methods that incorporate several proxy records in order to reconstruct spatially past climatic patterns (Riedwyl et al., 2009; Riedwyl et al., 2008; Touchan et al., 2005; Zhang et al., 2016). The CFR approaches use the spatial covariance calculated during overlapping calibration periods between the proxy series and the instrumental series. However, observational environmental records span at the most the past 150 years, not assuring that the covariance structure estimated at inter-annual timescales is also valid at decadal and multi-decadal timescales. The magnitude of the reconstruction uncertainties can vary depending on the different time scales (Briffa et al., 2001) and can be possibly underestimated if the proxy records have been previously screened according to their high covariance with instrumental records. As a large number of proxy records is usually screened, this high proxy-observation covariance could be due to chance (Osborn and Briffa, 2006). Another reason that could lead to an underestimation of the reconstruction uncertainties is the possible long term inhomogeneities existing in the proxy records (Jones et al., 2009).

The CFR methods can, however, be tested using climate simulations as a test bed, in experiments known as pseudo-proxy experiments, or PPEs (Gómez-Navarro et al., 2017; Werner et al., 2013). In the virtual world of a climate simulation all variables are known at all times. CFR methods can be recreated in this world by producing pseudo-proxies, using the simulated temperature at the grid-cell level and contaminating these records with statistical noise, so that the link between pseudo-proxies and grid-cell temperature resembles the observed link between proxy-records and instrumental temperature. In order to avoid possibly inconsistent results regarding the evaluation of the different CFR methods, correctly conducted and precisely described PPEs are necessary (Rutherford et al., 2013; Smerdon et al., 2010, 2013). A very important assumption underlying all CFR simulation studies relates to the general ability of the climate models to realistically simulate the spatial co-variance structure of the variable under consideration. This is difficult to assess, especially in the oceanic realm, which offers only a sparse observational network prior to 1980 (Reynolds et al., 2002). Taking into account the available re-analysis products, efforts can be undertaken to at least test the plausibility of state-of-the-art climate models in correctly simulating present-day ocean circulation and related quantities (Pyrina et al., 2017).

Given a realistic model simulation of the variable under consideration, the CFR methods can be then applied to the pseudo-proxy records in order to estimate a target variable, for instance the global, hemispheric or regional temperature (Mann and Rutherford, 2002; Mann et al., 2005; Moberg et al., 2008; Schenk and Zorita, 2012). The pseudo-reconstruction can be then compared to the simulated target variable. The differences between the pseudo-reconstructed and simulated target variable can give information about the deficiencies of the statistical CFR. PPEs also allow testing the sensitivity of the pseudo-reconstructions to changes in the density and location of the pseudo-proxy network (Annan and Hargreaves, 2012; Küttel et al., 2007; Smerdon, 2012).

The performance of CFRs has been tested using PPEs by a number of studies (Rutherford et al., 2003; Von Storch et al., 2004). Due to the significance of the large scale information estimated in CFRs, a growing number of studies have explicitly evaluated the spatial performance of CFRs (Dannenberg and Wise, 2013; Evans et al., 2014; Li and Smerdon, 2012; Smerdon et al., 2008; Wang et al., 2014). However, only a few studies have tested the differences in the spatial performance of CFR methods according to the modelled climate that is used as a test bed by the PPE (Mann et al., 2007; Smerdon et al., 2016; Smerdon et al., 2011). These studies tested the spatial skill of CFR methods using information from a composite proxy network including mostly terrestrial proxies. As CFRs depend on the characteristics of the proxy network used, such as proxy temporal resolution, growth season, character and level of noise (Christiansen and Ljungqvist, 2016), in our analysis we test CFR methods in the context of the annually resolved marine proxy network of *Arctica islandica*.

The potential of reconstructing the marine climate without the usage of terrestrial proxies is tested here for the first time using a network of annually resolved marine proxies over the NA basin. The performance of commonly applied CFR reconstruction methods is tested in different surrogate climates suitable for PPEs. A fundamental assumption for the construction of meaningful PPEs is that the spatiotemporal characteristics of the observed climate can be realistically simulated by the models that form the basis of the PPE. Therefore, the models that are used in this analysis were chosen based on their ability to simulate the spatiotemporal characteristics of the observed NA SSTs and have been previously evaluated in the context of paleoclimate reconstructions (Pyrina et al., 2017). The multivariate linear regression techniques tested are Principal Component Regression (PCR) and Canonical Correlation analysis (CCA). PCR has been previously used to reconstruct climate using only marine proxy records (Evans et al., 2002; Marchal et al., 2002), while both methods have been commonly applied in the context of CFRs using annually resolved proxies (Gómez-Navarro et al., 2015; Smerdon et al., 2016; Smerdon et al., 2011). Other linear methods widely used in the context of spatial CFRs include the RegEM algorithm of Schneider 2001 and the Analog Method, but none of those methods has been found to outperform the CFR techniques chosen to be tested in the current study (Christiansen et al., 2009; Gómez-Navarro et al., 2017; Smerdon et al., 2016; Smerdon et al., 2008).

## 2 Data and Methodology

We performed a PPE using modelled and reanalyzed gridded SSTs co-located with proxy sites of *Arctica islandica*. The grid point SSTs were taken from the Earth System Models (ESMs) CCSM4 and MPI-ESM-P, as well as the centennial in-situ observation-based estimate of SSTs, COBE2 (Hirahara et al., 2014). The SSTs are analysed for the summer period (June–August) and for the NA region between 60˚ W – 30˚ E and 40˚ N – 75˚ N, motivated by the growing season of *Arctica islandica* (Schöne et al., 2004; Schöne et al., 2005). The pseudo-proxy sample sites are based on five real world proxy sites of *Arctica islandica* including collection sites in the North Sea (NS: 1˚E, 58.5˚N) (Witbaard et al., 1997), the Irish Sea (IrS: 5˚W, 52.5˚N)

(Butler et al., 2009), the coast of Scotland (Sct: 7˚W, 56.5˚N) (Reynolds et al., 2013), the North Icelandic Shelf (IS: 20˚W, 66.5˚N) (Butler et al., 2013) and a location at Ingoya Island (InI: 24˚E, 71.5˚N) (Mette et al., 2015).

## 2.1 Data

### 2.1.1 Observational and Proxy data

We used the spatially interpolated reanalysis data set COBE2 (Hirahara et al., 2014). The COBE2 data set was developed by the Japanese Meteorological Agency, covers the period 1850–2013 AD and has a spatial resolution of 1˚x1˚ lat x lon. COBE2 data first pass quality control using combined a-priori thresholds and nearby observations and are later gridded using optimal interpolation. Data up to 1941 were bias-adjusted using "bucket correction", as the SST measurements were performed using a variety of buckets including canvas buckets and better insulated wooden or rubber buckets (Hirahara et al. 2014). The mean temperature change experienced by the water collected in a bucket until the temperature measurement is performed can be estimated and the temperature measurement can then be bias-corrected. COBE2 combines SST measurements from the release 2.0 of the International Comprehensive Ocean-Atmosphere Data Set, ICOADS (Worley et al., 2005), the Japanese Kobe collection, and readings from ships and buoys. The proxy data of *Arctica islandica* used in this analysis were downloaded from NOAA's (National Oceanic and Atmospheric Administration) National Centers for Environmental Information (NCEI, https://www.ncdc.noaa.gov/data-access/paleoclimatology-data/datasets) and refer to the 1357-year *Arctica islandica* reconstructed chronology from Butler et al., (2013). The multi-centennial absolutely dated chronology was reconstructed using annual growth increments in the shell of *Arctica islandica* and spans the period from 649 AD to 2005 AD.

### 2.1.2 Models

Two models are employed in this study, the CCSM4 model and the MPI-ESM-P model, which are part of the 5th phase of the Climate Model Intercomparison Project (CMIP5/ http://cmip-pcmdi.llnl.gov/cmip5/). The models' original output was re-processed and re-gridded to a regular grid for subsequent comparisons with the COBE2 data set. Therefore, the output was re-gridded onto a 1˚×1˚ horizontal resolution. The output of the models used in this study includes the combination of past 1000-year runs and historical runs (defined below), so that the period used spans from 850 AD to 1999 AD.

The CCSM4 (Gent et al., 2011) uses the atmosphere component Community Atmosphere Model, version 4 (CAM4) (Neale et al., 2013) and the land component Community Land Model, version 4 (CLM4) (Lawrence et al., 2012). Both components share the same horizontal grid (0.9˚ latitude × 1.25˚ longitude). The CCSM4 ocean component model (POP2) is based on the "Parallel Ocean Program", version 2 (Smith et al., 2010). The ocean grid has 320 × 384 points with nominally 1˚ resolution (~100 km) except near the equator, where the latitudinal resolution becomes finer to better simulate ENSO dynamics, as described in Danabasoglu et al., (2006). CICE4, the CCSM4 sea ice component model is based on version 4 of the Los Alamos National Laboratory "Community Ice Code" sea ice model (Hunke et al., 2008). The atmosphere, land, and sea ice components exchange both state information and fluxes through the coupler for every atmospheric time step. The fluxes between atmosphere and ocean are calculated in the coupler and communicated to the ocean component only once a day. The CCSM4 past 1000-year simulation covers the period from 850 AD to 1849 AD. The CCSM4 twentieth century simulations start from 1850 AD and end in December 2005 (Landrum et al., 2013). The forcings and boundary conditions follow the protocols of PMIP3 (Paleoclimate Modelling Intercomparison Project Phase III) (https://pmip3.lsce.ipsl.fr/wiki/doku.php/pmip3:design:lm:final) as discussed by Schmidt et al., (2012). The ice core based index of Gao et al., (2008) is used for the volcanic forcing, where several large volcanic eruptions have significantly larger aerosol optical depth compared to the Crowley et al., (2008) reconstruction for

PMIP3. Stratospheric aerosols are prescribed in three layers of the lower stratosphere in the model (Landrum et al., 2013). The Vieira et al., (2011) reconstruction merged to the Lean et al., (2005) reconstruction of Total Solar Irradiance (TSI) is used to prescribe TSI changes. The two TSI reconstructions are merged at 1834 in order to have a smooth transition from the CCSM past 1000-year runs to the twentieth-century simulations. For a seamlessly evolving land use change, the Pongratz et al., (2008) reconstruction of land use is merged with that of Hurtt et al., (2009) used in the CCSM4 twentieth-century simulations.

In MPI-ESM-P the atmosphere model ECHAM6 (Stevens et al., 2013) was integrated using a horizontal resolution of spectral truncation T63 (1.875°), while the ocean/sea-ice model MPIOM (Marsland et al., 2003) features a conformal mapping grid with nominal 1.5° resolution. MPI-ESM-P has a considerably higher spatial resolution in the NA (~15 km around Greenland), with one grid pole over Antarctica and one grid pole over Greenland. For land and vegetation the component JSBACH (Reick et al., 2013) is used and for the marine biogeochemistry the HAMOCC5 (Ilyina et al., 2013). The coupling at the interfaces between atmosphere and land processes, and between atmosphere and sea ice occurs at the atmospheric time step, which is also the time step of the land processes, except for the dynamic vegetation, which is updated once a year. The coupling between atmosphere and ocean as well as land and ocean occurs once a day. In the past 1000-year simulations a prescribed $CO_2$ forcing is used. For the effective radius and the optical depth of the volcanic aerosols the Crowley and Unterman reconstruction (Crowley and Unterman, 2013) is employed. The Pongratz et al., (2008) reconstruction is used for global land-cover and agricultural areas for both the past 1000-year runs and the historical simulations. The model was driven by the combined Vieira et al., (2011) and Wang et al., (2005) TSI reconstructions. In this work we used three experiments of the MPI-ESM model. The ensemble simulations r1 and r2 are started from the same initial conditions, but they consider a slightly different parameter setting regarding the volcanic aerosols. The experiments r2 and r3 consider the same parameter setting, but they differ in the initial conditions of the oceanic state (Jungclaus et al., 2014).

**2.2 Methodology**

Based on five proxy locations of *Arctica islandica* we reconstructed the summer SST evolution of the NA region, during the industrial (1850–1999 AD) period of the last millennium, using two CFR methods. In both approaches the goal is to reconstruct the spatially resolved SST fields (i.e. the predictand) using the grid point SSTs co-located with the real proxy locations (i.e. the predictors). To test the stationarity assumption of the calibration coefficients we repeat the reconstruction of the summer SSTs by calibrating our regression models for different calibration periods. The calibration periods include time spans during (a) the medieval period (1000-1049 AD), (b) the little ice age (1650-1699 AD), (c) the industrial period (1850-1999 AD), (d) the preindustrial period (850-1849 AD) and (e) recent years (1950-1999 AD). Although a real world reconstruction would be calibrated during an overlapping period between the proxy data and the instrumental data, in this study earlier calibration periods were also tested in order to be able to compare the model-based pseudoproxy results with the pseudoproxy results based on the reanalysis data. Furthermore, this approach gives us the opportunity to perform tests on calibration periods that are considered to be different in their climatic background state and judge the effect of these periods on the reconstruction skill.

The NA fields' inter-annual anomalies were reconstructed using the COBE2 reanalyzed SSTs and the CMIP5 modelled SSTs. To conclude whether the usage of different models or of reanalysis data has an effect on the reconstruction we correlated the original modelled or reanalyzed inter-annual detrended anomalies of the NA field with the reconstructed modelled or reanalyzed inter-annual detrended anomalies, respectively. In addition, we compared the temporal variances at grid-cell scale of the reconstructed and modelled SSTs. Other metrics used for the skill of the reconstruction at grid-cell scale were the Root Mean Square Error

190 (RMSE) of the reconstructed field relative to the target and the Reduction of Error (RE). The RMSE is a measure of both bias and differences in the variance between the reconstructed and the original SST evolution, with low values depicting low deviation from the original SST values (Wang et al., 2014). The RE measures the improvement of the fit of the reconstructed field against the original field, in comparison to the fit of the climatological mean of the calibration period against the original field (Cook et al., 1994).

### 2.2.1 Principal Component Regression

The first step of the PPE is the estimation of the NA SST field co-variances for the different calibration periods using Principal Component Analysis (PCA), (Eq. 1). Each eigenvector is represented by a spatial pattern (EOF, Empirical Orthogonal Function) which is associated with a temporal evolution (PC, Principal Component). In Eq. (1), $\vec{x}_t$ is the field vector of the NA SST anomalies and i the number of eigenvectors. In our analysis we kept the first 10 eigenvectors, as they represent more than 90% of variability and therefore most of the NA SST covariance is captured. The time, t, depends on the calibration period that we refer to.

$$\vec{x}_t = \sum_{i=1}^{10} PC_{i,t} \overrightarrow{EOF}_i, \tag{1}$$

For the five sampled SST time series ($Proxy_{j,t}$ with j representing the respective proxy location) we calibrated our regression model (Eq. 2) against the PCs determined during the calibration period, and estimated the calibration coefficients, $\alpha$, using PCR.

$$PC_{i,t} = \sum_{j=1}^{5} \alpha_{i,j} Proxy_{j,t} + \varepsilon \tag{2}$$

In our approach we assume that the PCs are linearly related to the pseudo-proxies, so that they represent large-scale climate variations for the SST fields. This relationship is modelled including a disturbance term or error variable, ε. The error could be an unobserved random variable that adds noise to the linear relationship between the dependent variable (PC) and the regressors (Proxy SSTs), and includes all effects on the regressors not related to the dependent variable (Christiansen, 2011). Based on PCR (Eq. 3) we then predict the principal components $\widehat{PC}_{i,t}$ during the reconstruction period, assuming that the calibration coefficients calculated in Eq. (2), are stationary in time. In this case the time, t, depends on the reconstruction period that we refer to.

$$\widehat{PC}_{i,t} = \sum_{j=1}^{5} \hat{\alpha}_{i,j} Proxy_{j,t} \tag{3}$$

Assuming that the dominant patterns of climate variability are similar in recent and past centuries, we predict the $\vec{x}_t$ field vector of the NA SST anomalies for the reconstruction period, using the predicted $\widehat{PC}_{i,t}$ and the $\overrightarrow{EOF}_i$ patterns calculated in Eq. (1). This stationarity assumption holds at least for multi-decadal timescales and allows us to deduce back in time the surface temperature patterns (Mann et al., 1998).

### 2.2.2 Canonical Correlation Analysis

The first step of the PPE using CCA is the eigenvalue decomposition and subsequent truncation of the NA SST field and the proxy SST field (Eq. 1), during the calibration interval. In this analysis the pseudo-proxy SST time series, or in other words the predictor data, are concatenated in a single SST field. After the transformation of the data to EOF coordinates we retained five empirical orthogonal functions, for each field, that were subsequently used to calculate the pairs of Canonical Correlation Patterns ($CCP^{NA}$, $CCP^{pr}$) and their time depended Canonical Coefficients ($CC^{NA}$, $CC^{pr}$), as shown in Eq. (4) and Eq. (5) for the NA SST field and the proxy SST field, respectively. The number of retained EOFs is equal to the maximum number of EOFs that can be kept in the case of the proxy field, as this number cannot exceed the number of proxy locations. For each estimated

CCP the correlation between the respective CCs is maximized and not correlated with the CCs of another pair of patterns. Therefore, the CCs fulfill the condition of orthogonality.

$$\vec{x}_t = \sum_{i=1}^{5} CC_{i,t}^{NA} \overrightarrow{CCP}_i^{NA} \tag{4}$$

$$\overrightarrow{Proxy}_t = \sum_{i=1}^{5} CC_{i,t}^{pr} \overrightarrow{CCP}_i^{pr} \tag{5}$$

The steps following are the same as the ones used in the method PCR, but instead of regressing and predicting the PCs we use the $CC_{i,t}^{NA}$. To reconstruct the $\vec{x}_t$ field vector of the NA SST anomalies we used the predicted $\widehat{CC}_{i,t}^{NA}$ and the $\overrightarrow{CCP}_i^{NA}$ patterns calculated in Eq. (4).

### 2.2.3 Noise addition

The PPEs were conducted using idealized pseudo-proxies and noise-contaminated pseudo-proxies. Idealized pseudo-proxies are the raw grid point SSTs, from simulations or reanalysis, co-located with the collection sites of the bivalve shell *Arctica islandica*, while in the case of the noise-contaminated pseudo-proxies the grid point temperatures are deteriorated by adding statistical noise in order to mimic more realistically the real world proxy records of *Arctica islandica*. The response of *Arctica islandica* to the SSTs, as of other bio-physiological proxies, might be non-stationary due to basic changes in ecosystem functioning or due to changes in factors such as food availability, salinity and turbidity. These processes are difficult to model statistically, as the non-stationarity may arise with very different character and has not been clearly characterized in real proxies. Regarding the level of non-climatic noise, it is usually not known in the real proxies and could be strongly dependent on the nature of the proxy indicator (von Storch et al., 2009).

The dynamics of many physical processes can be approximated by first or second order ordinary linear differential equations, whose discretized versions can be represented by autoregressive processes (Von Storch and Zwiers, 2001). An autoregressive processes of order k=0, where k is the time lag, is white noise ($Z_t$). An autoregressive processes of order k=1, or AR(1), represents a discretized first order linear differential equation and can be written as:

$$X_t = a_1 X_{t-1} + Z_t, \tag{6}$$

where $\alpha_1$ is the damping coefficient and $Z_t$ represents a random variable uncorrelated in time. The Yule-Walker equations (Von Storch and Zwiers, 2001) can be used to derive the first k+1 elements of the autocorrelation function and for an AR(1) process they give $\alpha_1 = \rho_1$, where $\rho_k$ is the autocorrelation for lag k. For a positive damping coefficient, an AR(1) process is unable to oscillate. Its 'spectral peak' is located at frequency $\omega = 0$ and therefore the variation $X_t$ behaves as red noise. Furthermore, the stationarity condition for an AR(1) process implies that $|\alpha_1| < 1$. Therefore, to approximate the variation of noise in *Arctica islandica* the autocorrelation function of an *Arctica islandica* chronology located at the Icelandic shelf was calculated and found equal to 0.4 at lag 1 year. It should be noted that an autocorrelation of 0.4 at lag 1 year is in the range of autocorrelation values found in tree ring-width chronologies (Büntgen et al., 2010; Larsen and MacDonald, 1995).

White noise series, $Z_t$, were generated in order to take into account the randomness of the noise. In a second step the white noise $Z_t$ series were transformed into red noise using the AR(1) process described in Eq. (6). The AR(1) noise series were finally re-scaled to the corresponding amplitude to achieve the desired relative variance of noise in the pseudoproxy record (see e.g. Smerdon et al., 2012). The pseudo proxy record is then composed of the sum of the simulated grid-cell temperature anomalies record and the AR(1) noise series.

## 3 Results

### 3.1 Ideal pseudo-proxies

In this section the results are based on five ideal pseudo-proxies co-located to *Arctica islandica* sites that are "sampled" from the
SST output fields of three realizations of the MPI-ESM model, the CCSM4 model and the COBE2 data set. The correlation
between the reconstructed and the original SST-anomaly evolution of the NA field is calculated for the industrial reconstruction
period and for the two different reconstruction methods. The methods are calibrated during the Medieval Period, the Little Ice
Age (LIA), the recent period, the industrial period and the preindustrial period, and the results are shown in the Appendix
(Figures 1S-5S). The results shown here regard the reconstruction of the industrial period when the regression models are
calibrated during the recent period and LIA (Fig. 1 and Fig. 2 respectively), and are shown for the CCSM4 model, the realization
r1 of the MPI-ESM model and the COBE2 data. Verification experiments were performed to test how well the calibration
coefficients work when the reconstruction interval is the same as the calibration interval; see Appendix Fig. 4S and Fig. 5S.
Moreover, the ratio of the estimated Standard Deviation (SD) according to the reconstructed SST-anomaly evolution of the NA
field and the estimated SD according to the original SST-anomaly evolution was calculated ($SD_{reconstruction}/SD_{original}$) and shown in
Fig. 1 and Fig. 2 for the recent and LIA calibration periods, respectively.

Looking at the results of the same model simulation and method but for different calibration periods we can see that the spatial
skill of the reconstruction does not significantly change according to the calibration period of the regression models. Therefore,
the calibration coefficients can be considered stationary. Additionally, the SST patterns shown by the correlation maps do not
profoundly change when a different method is applied, indicating that with both methods the SST evolution of the eastern NA
basin can be reconstructed using only five proxy locations of *Arctica islandica*. More pronounced differences are found between
the pseudo-reconstructions conducted with each model and also with the pseudo-reconstruction using the reanalysis data. For
instance, when the industrial period is reconstructed using the recent calibration period (Fig. 1) according to CCSM4 and
COBE2, the SSTs between Iceland and Norway can be skilfully reconstructed with a correlation coefficient approximately equal
to r≈+0.8, but according to MPI-ESM the skill drops for all realizations (Appendix, Fig. 3S). Another interesting finding is that
the SSTs along the south-east coast of Greenland can be reconstructed with a good skill using the CCSM4 model (r≈+0.9), while
the reconstruction skill drops for the COBE2 data (r≈+0.7) and the MPI-ESM model (r≈+0.5). The correlation maps provided
from the CCSM4 model and the COBE2 data are more homogenous than the ones given by the MPI-ESM-P model. The areas
that exhibit correlation values higher than r≈+0.6 are more widespread in CCSM4 and COBE2 compared to MPI-ESM-P.

The standard deviation ratios indicate that variance is mostly preserved in areas where field correlations are high. According to
models and the COBE2 reanalysis data, both CFR methods result in a variance loss in most of the NA basin, with the largest
decrease occurring over the West Atlantic basin (Appendix, Fig. 8S). The RMSE of the reconstructed SST evolution obtains
generally low values close to the proxy sites for both methods and for different calibration periods, indicating that in those
regions the prediction capacity is better (Appendix, Fig. 10S and Fig. 11S). The RE patterns (Appendix, Fig. 12S and Fig. 13S)
follow similar spatial patterns to the ones shown by the correlation maps (Appendix, Fig. 3S and Fig. 2S), depicting higher
values over the eastern Atlantic basin especially for the PCR method.

## 3.2 Noise contaminated pseudo-proxies

The results of this section are illustrated as in section 3.1, but the calculations were performed with noise-contaminated pseudo-proxies. The methods calibrated during the recent period and LIA are shown in Fig. 3 and Fig. 4, respectively, for the CCSM4 model, the realization r1 of the MPI-ESM model and the COBE2 data. Compared to the results regarding the ideal pseudo-proxies, the reconstruction skill has decreased with the maximum correlation between the reconstructed and the original SST-anomaly evolution of the proxy sites reaching r=+0.8, depending on the model used and the pseudo-proxy location (Appendix Figs. 1S-5S). The areas that exhibit correlation with values higher than r=+0.6 are more limited in extent when compared to the idealized PPE, being mostly the regions around the British Isles, as shown by COBE2 and the MPI-ESM model (Fig. 3). According to both methods, the eastern part of the NA can be skilfully reconstructed with the proxy sites of *Arctica islandica*. The differences amongst models are more profound than the differences amongst the different calibration periods.

Whilst in most cases the results regarding the reconstruction of the eastern Atlantic basin do not profoundly change according to the reconstruction method and the calibration period, the realization r2 of the MPI-ESM model shows that those aspects might play an important role. When we compare the results of the realization r2 of the MPI-ESM as calculated with the CCA method during the LIA calibration period (Appendix, Fig. 2S) to the respective results calculated during the rest of the calibration periods (Appendix Figs. 1S-5S), the skill of the reconstruction of the east Atlantic basin appears generally lower in the Appendix Fig. 2S. Moreover, for the realizations r3 and r2 and for the LIA calibration period (Appendix, Fig. 2S) there is an anti-correlation of around r~-0.3 over the West Atlantic, an aspect shown for the realization r2 also in the ideal PPE. These results indicate that the CCA method would be problematic regarding the reconstruction of the NA SSTs over those areas, based on the r2 and r3 realizations of the MPI-ESM model, as an anti-correlation between the original and the reconstructed temperature evolution has no physical meaning. According to the noise contaminated experiment (Fig. 3 and Fig. 4), the standard deviation ratios indicate that the CFRs conducted using either the PCR or the CCA method are afflicted with a loss in variance over the whole NA basin. Areas where field correlations are high, exhibit the highest values of RE (~0.5, see Appendix; Fig. 12S and Fig. 13S) and generally low values of RMSE (Appendix, Fig. 10 and Fig. 11).

The results of this section were also calculated with the PCR and CCA methods, but for the contamination of the pseudo-proxies an additional realization of the AR(1) model (Eq. 6) was used. The results are given in the Appendix (Fig. 14S, Fig. 15S, Fig. 16S and Fig. 17S). This analysis confirms the conclusions drawn by the CFRs using the initial AR(1) noise realization, for all reconstruction skill metrics, as well for the importance of the model simulation in comparison to the calibration period and the reconstruction method.

## 3.3 Test of proxy locations

We also tested the contribution of the different *Arctica islandica* proxy sites on the reconstruction skill. We reconstructed the NA SSTs during the industrial period and calibrated our regression models during the recent period, according to (a) the sites on the Icelandic Shelf and North Sea (Figures with NP=2) and (b) the sites on the Icelandic Shelf, North Sea and Ingoya Island (Figures with NP=3). The reconstruction is performed with both statistical methods, using ideal proxies (Appendix, Fig. 6S) and noise contaminated proxies (Appendix, Fig. 7S). In Fig. 5 the results using three *Arctica islandica* sites are shown for both ideal proxies and noise contaminated proxies.

Regarding the ideal PPE (Appendix, Fig. 6S), only the results of the r2 realization of the MPI-ESM-P model indicate that the PCR method leads to a greater reconstruction skill than the respective results according to the CCA method. That result applies for both two and three proxy locations. Moreover, anti-correlation between the reconstructed and original SST evolution occurs with the CCA method, mostly when only two proxy sites are used. Additionally, the differences amongst models and amongst models and data are more profound for the CCA method than for the PCR, because the number of CCPs that can be used for the CFR is limited by the number of proxies used (two CCPs for NP=2 and three CCPs for NP=3). Comparing the results of Fig. 5, regarding the reconstruction using three *Arctica islandica* sites to the corresponding results of Fig. 1, which are based on the reconstruction using five *Arctica islandica* sites, we observe that in most cases the sites in the Irish Sea and the off the coast of Scotland not only contribute to the reconstruction of the SSTs on their surrounding waters but they additionally increase the reconstruction skill over the east Atlantic basin.

Regarding the noise-contaminated PPE (Appendix, Fig. 7S), the results indicate that the PCR method leads to a greater reconstruction skill in the case of two proxy locations, but none of the methods has beneficial characteristics in the case of three proxy locations. Generally, when the CCA method is used regions of anti-correlation are apparent in the pseudo-proxy results of the r1 and r3 realizations of the MPI-ESM model and of the COBE2 data. Comparing the Appendix Fig. 7S to the Appendix Fig. 3S for the contaminated experiment, we see that the addition of the sites in the Irish Sea and the coast of Scotland also results in an increase of the reconstruction skill of the east Atlantic basin for all models and data, and for both methods. The reason that the sites in the Irish Sea and the coast of Scotland increase the overall reconstruction skill is that the oceanographic variability of the western British Isles is dominated by the northward transport of warm saline waters advected by the North Atlantic Current (Inall et al., 2009).

## 4 Discussion

### 4.1 Stationarity of the calibration coefficients

Both CFR methods showed that the spatial skill of the reconstruction does not profoundly change according to the period during which the regression models were calibrated, even when the calibration period chosen relates to the recent period, a period dominated by anthropogenic forcing. This result is robust amongst models and amongst models and data and could be an indication that the NA basin leading modes of SST variability remain stationary during several periods of the last millennium for different climatic background states. However, changes in teleconnections on multi-decadal to centennial time scales in control model simulations or externally forced simulations have been proposed by several studies (e.g. Gallant et al., 2013; Müller and Roeckner, 2008). Regarding the NA basin over the last half millennium, non-stationary AMO-like fluctuations were found in proxy-data and climate model simulations (Enfield and Cid-Serrano, 2006). Apparent non-stationarity in multi-decadal to centennial AMO variability was also found in a controlled climate simulation covering multi-millennial time scales (Zanchettin et al., 2010). In contrast to the previous results mentioned, Zanchettin et al., (2013) found that the simulated leading modes of SST variability for the extra-tropical NA basin during the period 800–2005 AD, are similar to the simulated and observed regional SST variability patterns over the last 160 years. At inter-annual time scales, and for each calibration period used in this study, we calculated the teleconnections of the NA SSTs to the regions co-located to the *Arctica islandica* sites, using the COBE2 reanalysis data and the output of the MPI-ESM and CCSM4 model. Generally, models and reanalysis data show that the regions that exhibit high and statistically significant correlations ($r \geq +0.8$) exist mainly between the proxy sites and their surrounding waters and that these teleconnections are stable in time (see in Appendix; Fig. 18S-Fig22S).

Another reason that could potentially explain the similarities of the reconstruction skill amongst different calibration periods could be that the relationship of the SSTs of the regions where the proxies are located and the modes of variability that influence the NA basin remain unchanged. Lohmann et al., (2005) found that during the period 1900–1998 AD the Arctic Oscillation-related temperature teleconnections, show weak decadal variations in some regions of the NA. The Arctic Oscillation signature

in climate variables was detectable also during the spring season, which is of practical relevance as the climate information obtained from most terrestrial and marine proxy archives is more linked to the growing season rather than to winter (Cook et al., 2002). The NA regions that Lohmann et al., (2005) identified with stable teleconnections include almost all the proxy locations used in our study (see in Lohmann et al., 2005; Fig. 4), but their analysis regards only the recent period and decadal variations. Regarding the North Atlantic Oscillation (NAO), variable teleconnections have been detected in coupled ocean-atmosphere

model simulations (Raible et al., 2014; Raible et al., 2001; Trouet et al., 2009; Vicente-Serrano and López-Moreno, 2008; Zorita and González-Rouco, 2002). Moreover, model inaccuracies in the representation of the mean NA climate, NAO and other modes of climate variability that influence the NA SSTs could lead to a poor representation of the NA regions. However, as CFRs are covariance-based approaches, a poor representation of large scale climatic patterns that describe spatial relationships between different regions of the NA basin would lead to a generally low reconstruction skill in the NA basin, which is not consistent with

the results of this study. Therefore, a more plausible reason that explains the similarities of the reconstruction skill amongst different calibration periods relates to the minimization of possibly existing non-stationarities by using proxy sites from multiple regions (Batehup et al., 2015).

## 4.2 Model and method dependent spatial skill

As presented above, more profound spatial differences in the reconstruction skill are found for the different model simulations,

rather than for the different periods in the last millennium that were used to calibrate the regression models. The pseudo-proxy results of the CCSM4 model are found to be closer to the pseudo-proxy results of the COBE2 data set, than the ones calculated based on the MPI-ESM-P model. Concerning EOFs for the summer means during the period 1950-1999 AD for the NA, Pyrina et al., (2017) found that the leading patterns of SST variability as given by the COBE2 show more similarities to the EOFs calculated by the CCSM4 model than to the ones calculated by the MPI-ESM-P model. Moreover, the teleconnection patterns of

the Icelandic Shelf and North Sea proxy sites to the SSTs of the NA basin were evaluated and the CCSM4 model was again found to be closer to the teleconnection maps shown by COBE2 in terms of both resemblance of magnitude and spatial patterns. Therefore, as the large scale SST variability of NA basin of the CCSM4 model was found to be closer to the observational data, this generally explains the resemblance of the COBE2 reconstruction skill by the CCSM4 model. Another interesting finding relates to the fact that the SSTs on the path of the east Greenland current can be reconstructed with a good skill only with the

CCSM4 model, and this is one of the points where the CCSM4 model does not agree with the COBE2 data (Fig. 1, Fig. 3). The reason for this disagreement could be due to an unrealistic artefact of the COBE2 data set, as there is a suspect abrupt jump in the SSTs of the regions north of Iceland in the COBE2 dataset (see in Loder et al. 2015; their Fig. 7).

The usage of MPI-ESM-P leads to more heterogeneous correlation maps. Even though the ocean component of the CCSM4 model has a higher spatial resolution over most regions than the MPI-ESM-P, in the NA Ocean the ocean component of the MPI-

ESM-P model has considerably higher spatial resolution on the original curvilinear model grid. That technical characteristic could explain the more heterogeneous results of the MPI-ESM-P in the NA area. Differences between the two models could also arise from the slightly different volcanic forcing used. Booth et al., 2012 found that aerosol concentration changes influence the

simulated spatial response of the SSTs. The effect on the reconstruction skill due to a change in the standard deviation of the volcanic aerosol forcing can be seen by the comparison between the ensemble members r1 and r2 of the MPI-ESM model (e.g. lower reconstruction skill in r2 along the coast of Norway; Appendix Fig. 2S, Fig. 16S). These members are integrated with the same model version, started with the same ocean state, but with volcanic aerosol size equal to 1.2 μm in r1 and 1.8 μm in r2. In addition, differences between the two models could also arise from the implementation of the aerosol component which is prescribed in the MPI-ESM model and interactive in the CCSM4. Rotstayn et al., (2010) found an improvement of the Australian mean seasonal climate by including in the CSIRO model an interactive aerosol scheme.

The differences of reconstruction skill according to CCA or PCR can be better identified in the case of the MPI-ESM model, as indicated by the spatially more heterogeneous correlations between the original and the reconstructed SST anomalies. For the reconstruction of the NA SSTs using the PCR method, we select for calibration the dominant patterns of SST variability of the NA basin without simultaneous consideration of the patterns of the proxy data, while with the usage of the CCA method we exploit only those patterns with time histories related to the time histories of the proxy network. However, both methods share similar results in all reconstruction skill metrics evaluated in this study. Differences in the spatial skill of the methods can be better seen when noise contaminated pseudo-proxies are used, as in this case the loss of information is greater, because the noise contaminated pseudo-proxies do not explain 100% of the SST signal.

Regarding both the ideal and the noise-contaminated experiment, the reconstruction methods evaluated herein lead to a loss in variance. The decrease in variance can be expected for CFR methods that are based on linear regression and combine the signal of the targeted variable with unrelated variability (Smerdon et al., 2011; von Storch et al., 2004). As described in Christiansen, (2011) when the proxy is chosen as the independent variable, then the equation error results in underestimations of the magnitude of the predicted values. In the case of ideal pseudo-proxies a proportion of variance is lost due to the disturbance term (Eq. 2) that is not modelled during the reconstruction (Eq. 3), while in the case of noise-contaminated pseudoproxies additional variance loss is expected due to the non-perfect correlation between the pseudoproxy and the local climate variable (McCarroll et al., 2015). Moreover, the truncation of the eigenvalue spectra by the PCR and CCA methods could lead to additional variance losses (Smerdon et al., 2010).

## 5 Conclusions

The tested CFR methods can have an impact on the spatial skill of the reconstruction, especially when noise contaminated pseudo-proxies are used. Therefore, it is important to assess CFR methods with noise-contaminated proxies rather than with ideal noise-free ones. Moreover, regardless of the model simulation used, both CFRs methods are affected by loss of reconstruction variance. Field correlations are high on the east Atlantic basin where less variance loss is observed and a better prediction skill is indicated by the RMSE and RE. The CCA method is problematic when a significantly low number of proxies is used (two and three proxies), but the spatial skill of the reconstruction using CCA and five proxy locations is similar to the results calculated with the PCR method. Even though the models used as the basis of the PPEs were previously evaluated in the context of CFRs, it is found here that the most important differences pertaining the spatial skill of the reconstruction are caused by the choice of model simulation used, rather than by the specific CFR method or by the calibration period. With our PPEs we have demonstrated that the SSTs can be skilfully reconstructed not only around the proxy sites of *Arctica islandica*, but in a broader area of the eastern Atlantic basin. This is an important result in the context of spatially resolved CFRs in the NA basin, which

lacks annually resolved marine proxy data, as a small sized proxy network of the marine bivalve mollusk *Arctica islandica* (five proxy locations) could provide valuable information for SST reconstructions of the northeast Atlantic Ocean.

**Author Contributions**

455 The analysis was performed by M. Pyrina with the consultation of S. Wagner and E. Zorita. M. Pyrina prepared the manuscript with contributions of co-authors.

**Acknowledgements**

The work was carried out in the framework of the European Initial Marie Curie Training network ARAMACC (Annually resolved Archives of Marine Climate Change). This project has received funding from the European Union's Seventh 460 Framework Programme for research, technological development and demonstration under grant agreement no 604802. The authors thank the CCSM4 and MPI-ESM modelling groups participating in the CMIP5 initiative for providing their data and the authors of the COBE2 data set for making their data available. Also, the authors would like to explicitly thank the three anonymous reviewers who helped to improve the manuscript's technical and scientific content.

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

Zorita, E. and González-Rouco, F.: Are temperature-sensitive proxies adequate for North Atlantic Oscillation reconstructions?, Geophysical research letters, 29, 2002.

**Figure Captions**

Figure 1: Correlation coefficient (columns 1 and 3) and SD ratio (columns 2 and 4) between the reconstructed and the original SST-anomaly evolution of the NA field during the industrial era. The results regard the ideal experiment and are given for the recent calibration period (1950—1999 AD) and for the two different reconstruction methods (1st and 2nd column CCA, 3rd and 4th column PCR) for the models CCSM4 (first row) and MPI-ESM-P r1 (second row) and the reanalysis data COBE2 (third row).

Figure 2: Correlation coefficient (columns 1 and 3) and SD ratio (columns 2 and 4) between the reconstructed and the original SST-anomaly evolution of the NA field during the industrial era. The results regard the ideal experiment and are given for the LIA calibration period (1650—

1699 AD) and for the two different reconstruction methods (1st and 2nd column CCA, 3rd and 4th column PCR) for the models CCSM4 (first row) and MPI-ESM-P r1 (second row).

Figure 3: Same as in Fig. 1, but for the noise-contaminated pseudo-proxies.

Figure 4: Same as in Fig. 2, but for the noise-contaminated pseudo-proxies.

Figure 5: Correlation coefficient between the reconstructed and the original SST-anomaly evolution of the NA field during the industrial era for the ideal (columns 1 and 3) and the noise contaminated experiment (columns 2 and 4) when the number of proxy locations used is NP=3. The results are given for the recent calibration period (1950—1999 AD), for two different reconstruction methods (1st and 2nd column CCA, 3rd and 4th column PCR) and for the models CCSM4 (first row) and MPI-ESM-P r1 (second row), and the reanalysis data COBE2 (third row).

**APPENDIX figure captions**

Figure 1S: Correlation coefficient between the reconstructed and the original SST-anomaly evolution of the NA field for the ideal (column 1 and 3) and the noise contaminated experiment (column 2 and 4). The results are given for the MCA calibration period (1000—1049 AD) and for the two different reconstruction methods (1st and 2nd column CCA, 3rd and 4th column PCR) for the models CCSM4 (1st row) and three realizations of the MPI-ESM-P model (2nd, 3rd and 4th row).

Figure 2S: As in Fig. 1S, but the results are given for the LIA calibration period (1650—1699 AD).

Figure 3S: As in Fig. 1S, but the results are given for the recent calibration period (1950—1999 AD). The last row contains the pseudo-proxy results of the COBE2 reanalysis data.

Figure 4S: As in Fig. 3S, but the results are given for the industrial calibration period (1850—1999 AD).

Figure 5S: As in Fig. 1S, but the results are given for the pre-industrial calibration period (850—1849 AD).

Figure 6S: Correlation coefficient between the reconstructed and the original SST-anomaly evolution of the NA field during the industrial era, when the number of proxy locations used is NP=2 (1st and 3rd column) and NP=3 (2nd and 4th column). The results are given for the recent calibration period (1950—1999 AD) and for the two different reconstruction methods (1st and 2nd column CCA, 3rd and 4th column PCR) for the model CCSM4 (1st row), three realizations of the MPI-ESM-P model (2nd, 3rd and 4th row) and the COBE2 reanalysis data (5th row).

Figure 7S: As in Fig. 6S, but the results are given for the noise contaminated pseudo-proxy experiment.

Figure 8S: SD ratio between the reconstructed and the original SST-anomaly evolution of the NA field during the industrial period, for the ideal (column 1 and 2) and the noise contaminated (column 2 and 4) pseudo-proxy experiment. The results are given for the recent calibration and for the two different reconstruction methods (1st and 2n column CCA, 3rd and 4th column PCR) for the models CCSM4 (1st row) and three realizations of the MPI-ESM-P model (2nd, 3rd and 4th row), as well as the COBE2 data (5th row).

Figure 9S: As in Fig. 8S, but for the LIA calibration period and for the model CCSM4 (1st row) and three realizations of the MPI-ESM-P model (2nd, 3rd and 4th row).

Figure 10S: RMSE of the reconstructed SST-anomalies of the NA field during the industrial period, for the ideal (column 1 and 3) and the noise contaminated (column 2 and 4) pseudo-proxy experiment. The results are given for the recent calibration and for the two different reconstruction methods (1st and 2nd column CCA, 3rd and 4th column PCR) for the models CCSM4 (1st row) and three realizations of the MPI-ESM-P model (2nd, 3rd and 4th row), as well as the COBE2 data (5th row).

Figure 11S: As in Fig. 10S, but for the LIA calibration period and for the model CCSM4 (1st row) and three realizations of the MPI-ESM-P model (2nd, 3rd and 4th row).

Figure 12S: RE of the reconstructed SST-anomalies of the NA field during the industrial period, for the ideal (column 1 and 3) and the noise contaminated (column 2 and 4) pseudo-proxy experiment. The results are given for the recent calibration and for the two different reconstruction methods (1st and 2nd column CCA, 3rd and 4th column PCR) for the models CCSM4 (1st row) and three realizations of the MPI-ESM-P model (2nd, 3rd and 4th row), as well as the COBE2 data (5th row).

Figure 13S: As in Fig. 12S, but for the LIA calibration period and for the model CCSM4 (1st row) and three realizations of the MPI-ESM-P model (2nd, 3rd and 4th row).

Figure 14S: The correlation and SD ratio between the reconstructed and the original SST-anomaly evolution of the NA field during the industrial period are given in columns 1 and 2, respectively. The RMSE and RE of the reconstructed SST anomalies are shown in columns 3 and 4 respectively. The results are calculated using CCA, regard a second noise realization of the pseudo-proxy experiment and are given for the recent calibration period and for the models CCSM4 (1st row) and three realizations of the MPI-ESM-P model (2nd, 3rd and 4th row), as well as the COBE2 data (5th row).

Figure 15S: As in Figure 14S, but the results are calculated using PCR.

Figure 16S: The correlation and SD ratio between the reconstructed and the original SST-anomaly evolution of the NA field during the industrial period are given in columns 1 and 2, respectively. The RMSE and RE of the reconstructed SST anomalies are shown in columns 3 and 4 respectively. The results are calculated using CCA, regard a second noise realization of the pseudo-proxy experiment and are given for the LIA calibration period and for the models CCSM4 (1st row) and three realizations of the MPI-ESM-P model (2nd, 3rd and 4th row), as well as the COBE2 data (5th row).

Figure 17S: As in Fig. 16S, but the results are calculated using PCR.

Figure 18S: One point correlation maps between the SSTs co-located to the IS site and the NA basin, for the MCA period (1st column), the LIA period (2nd column), the industrial period (3rd column) and the recent period (4th column). Hatched areas indicate statistical significance on the 99% level.

Figure 19S: As in Figure 18S, but for the NS site. Hatched areas indicate statistical significance on the 99% level.

Figure 20S: As in Figure 18S, but for the Sct site. Hatched areas indicate statistical significance on the 99% level.

Figure 21S: As in Figure 18S, but for the IrS site. Hatched areas indicate statistical significance on the 99% level.

Figure 22S: As in Figure 18S, but for the InI site. Hatched areas indicate statistical significance on the 99% level.

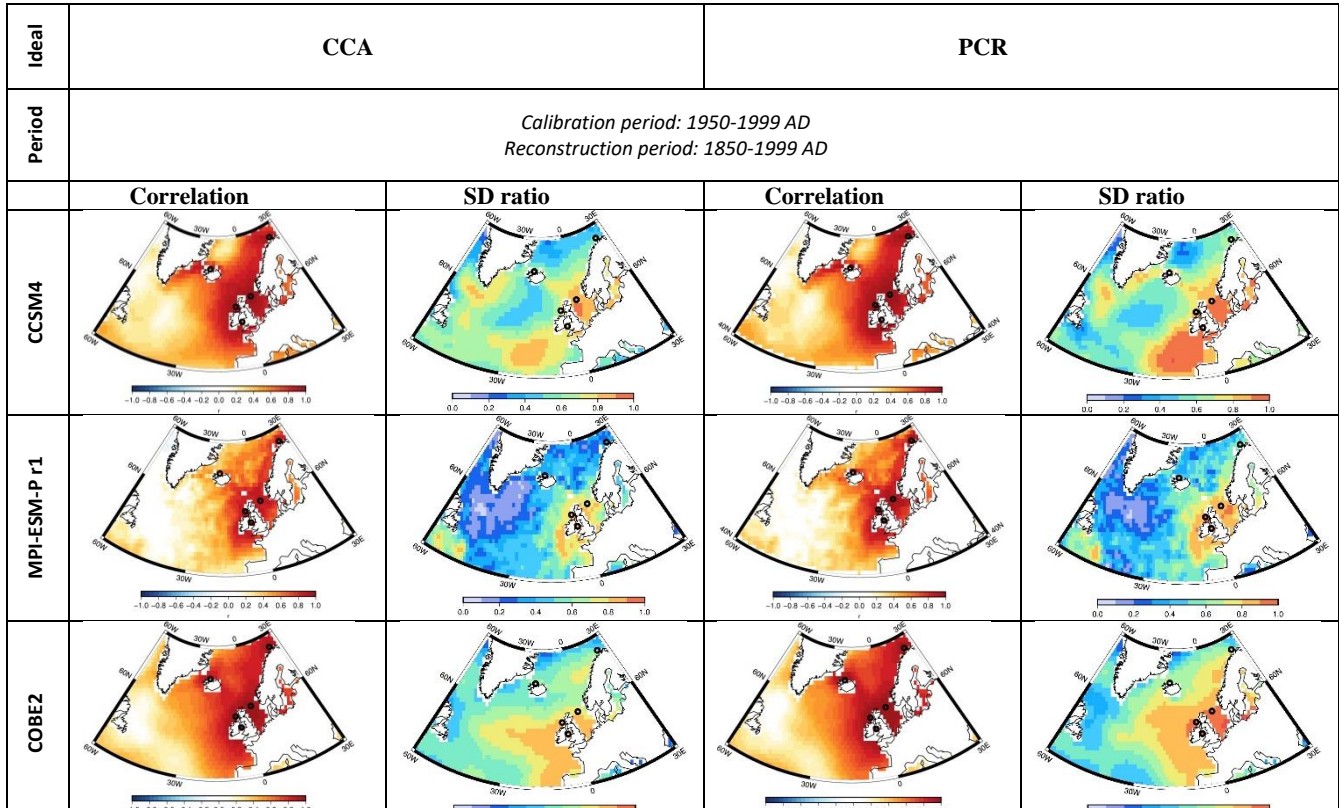

**Figure 1: Correlation coefficient (columns 1 and 3) and SD ratio (columns 2 and 4) between the reconstructed and the original SST-anomaly evolution of the NA field during the industrial era. The results are given for the recent calibration period (1950—1999 AD) and for the two different reconstruction methods (1st and 2nd column CCA, 3rd and 4th column PCR) for the models CCSM4 (first row) and MPI-ESM-P r1 (second row) and the reanalysis data COBE2 (third row).**

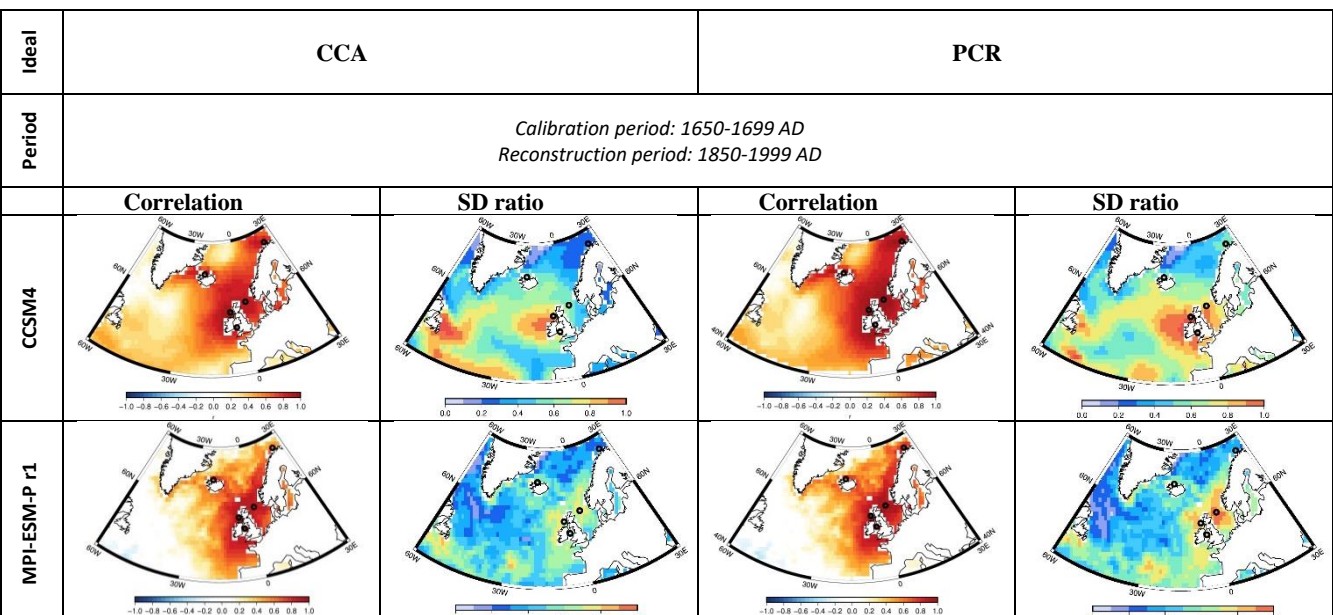

**Figure 2: Correlation coefficient (columns 1 and 3) and SD ratio (columns 2 and 4) between the reconstructed and the original SST-anomaly evolution of the NA field during the industrial era. The results are given for the LIA calibration period (1650—1699 AD) and for the two different reconstruction methods (1st and 2nd column CCA, 3rd and 4th column PCR) for the models CCSM4 (first row) and MPI-ESM-P r1 (second row).**

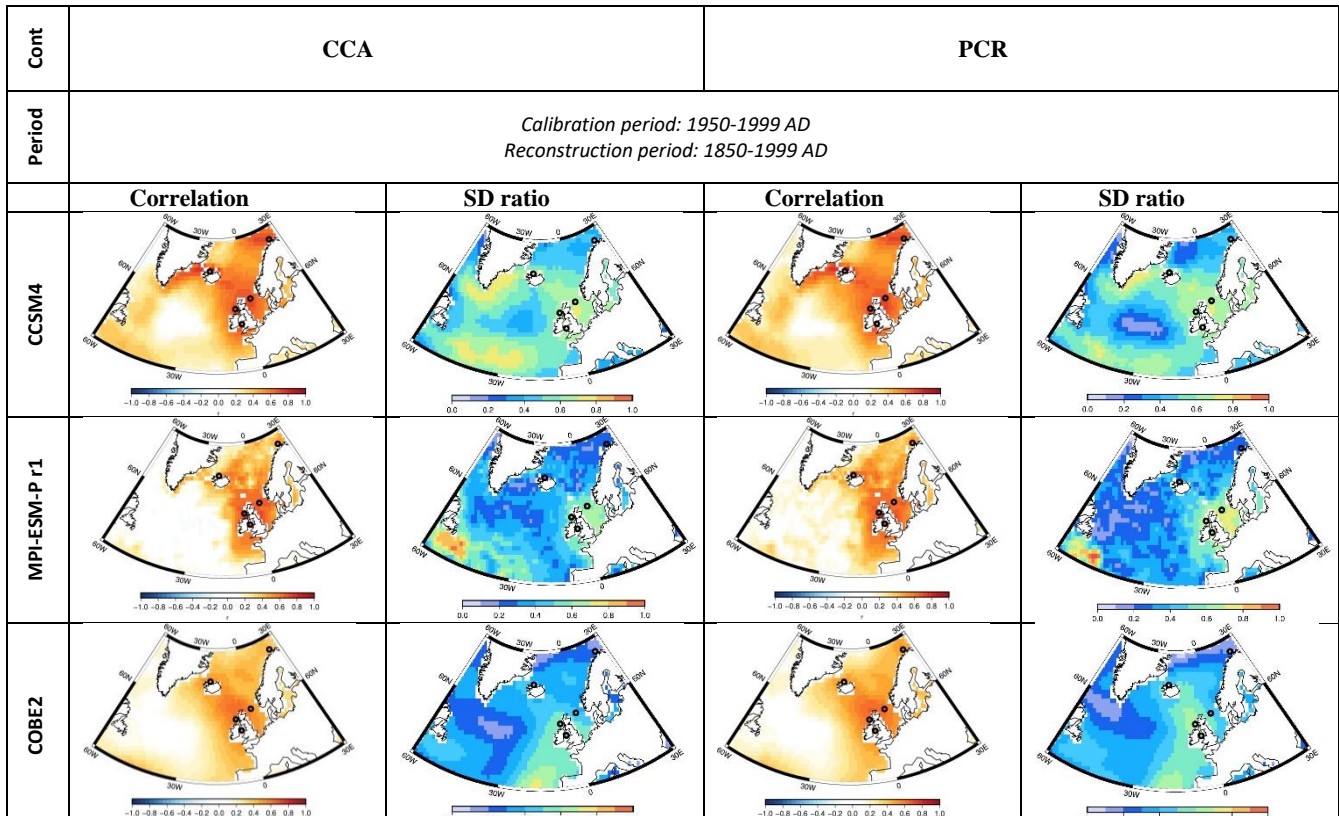

**Figure 3: Same as in Fig. 1, but for the noise-contaminated pseudo-proxies.**

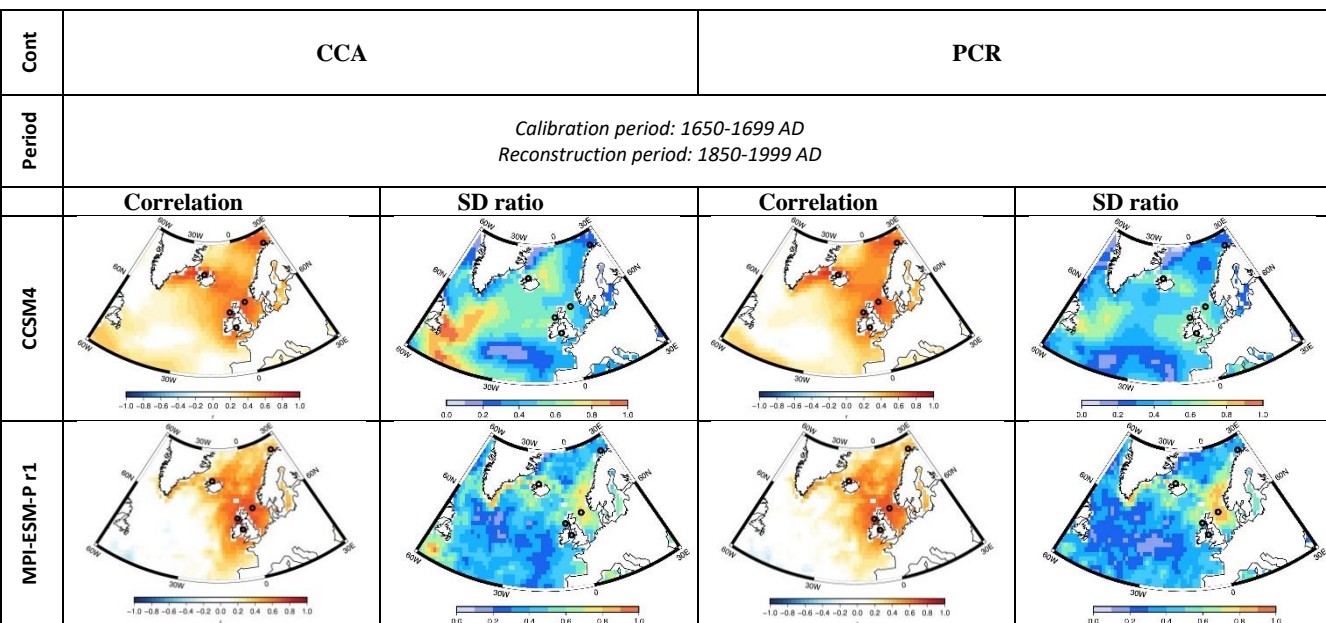

**Figure 4: Same as in Fig. 2, but for the noise-contaminated pseudo-proxies.**

| Method | CCA | | PCR | |
|---|---|---|---|---|
| Period | Calibration period: 1950-1999 AD | | | |
| | Reconstruction period: 1850-1999 AD | | | |
| NP | Number of Pseudo-proxies = 3 | | | |
| Experiment | **Ideal** | **Contaminated** | **Ideal** | **Contaminated** |
| CCSM4 | | | | |
| MPI-ESM-P r1 | | | | |
| COBE2 | | | | |

**Figure 5: Correlation coefficient between the reconstructed and the original SST-anomaly evolution of the NA field during the industrial era for the ideal (columns 1 and 3) and the noise contaminated experiment (columns 2 and 4) when the number of proxy locations used is NP=3. The results are given for the recent calibration period (1950—1999 AD), for two different reconstruction methods (1st and 2nd column CCA, 3rd and 4th column PCR) and for the models CCSM4 (first row) and MPI-ESM-P r1 (second row), and the reanalysis data COBE2 (third row).**

# APPENDIX

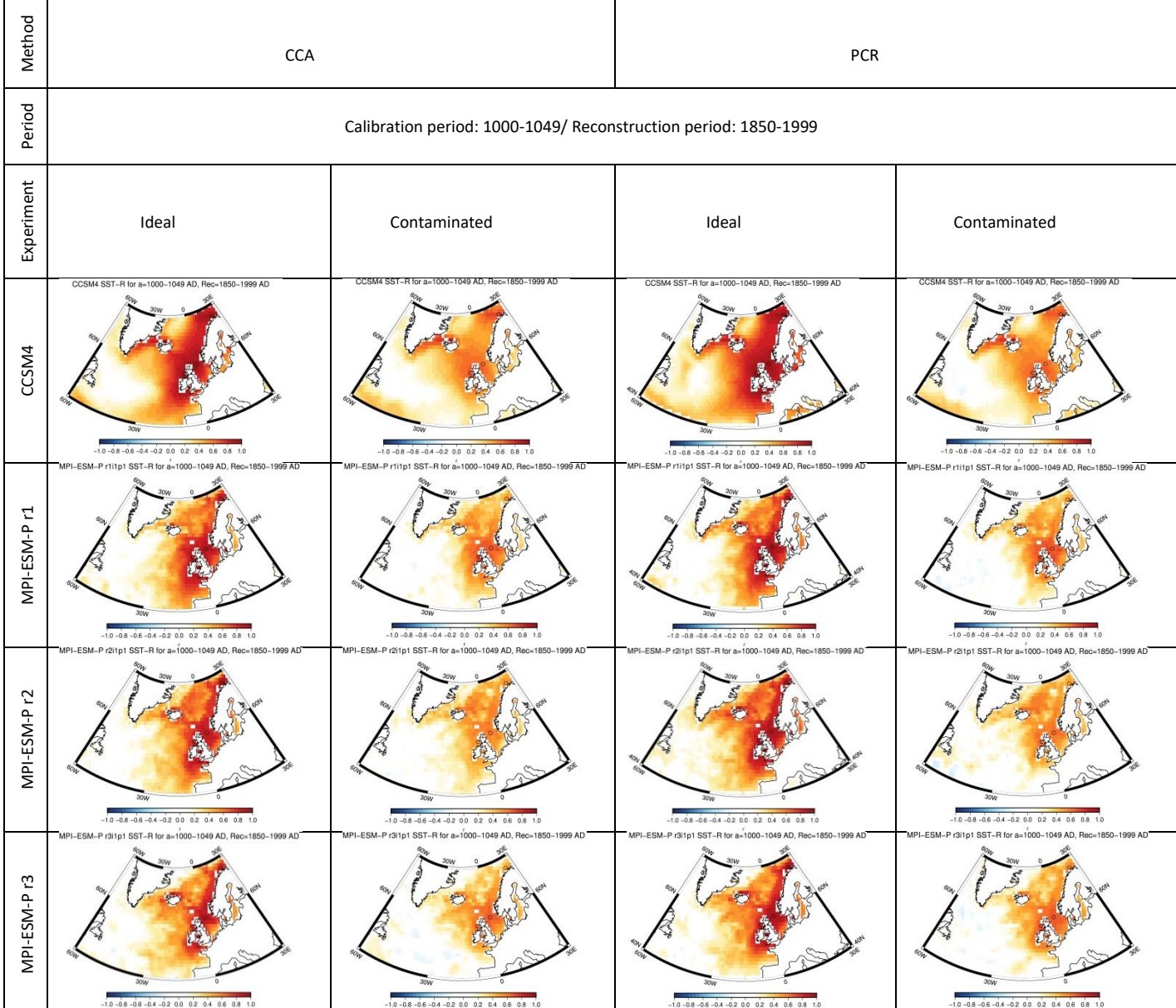

Figure 1S: Correlation coefficient between the reconstructed and the original SST-anomaly evolution of the NA field for the ideal (column 1 and 3) and the noise contaminated experiment (column 2 and 4). The results are given for the MCA calibration period (1000—1049 AD) and for the two different reconstruction methods (1st and 2nd column CCA, 3rd and 4th column PCR) for the models CCSM4 (1st row) and three realizations of the MPI-ESM-P model (2nd, 3rd and 4th row).


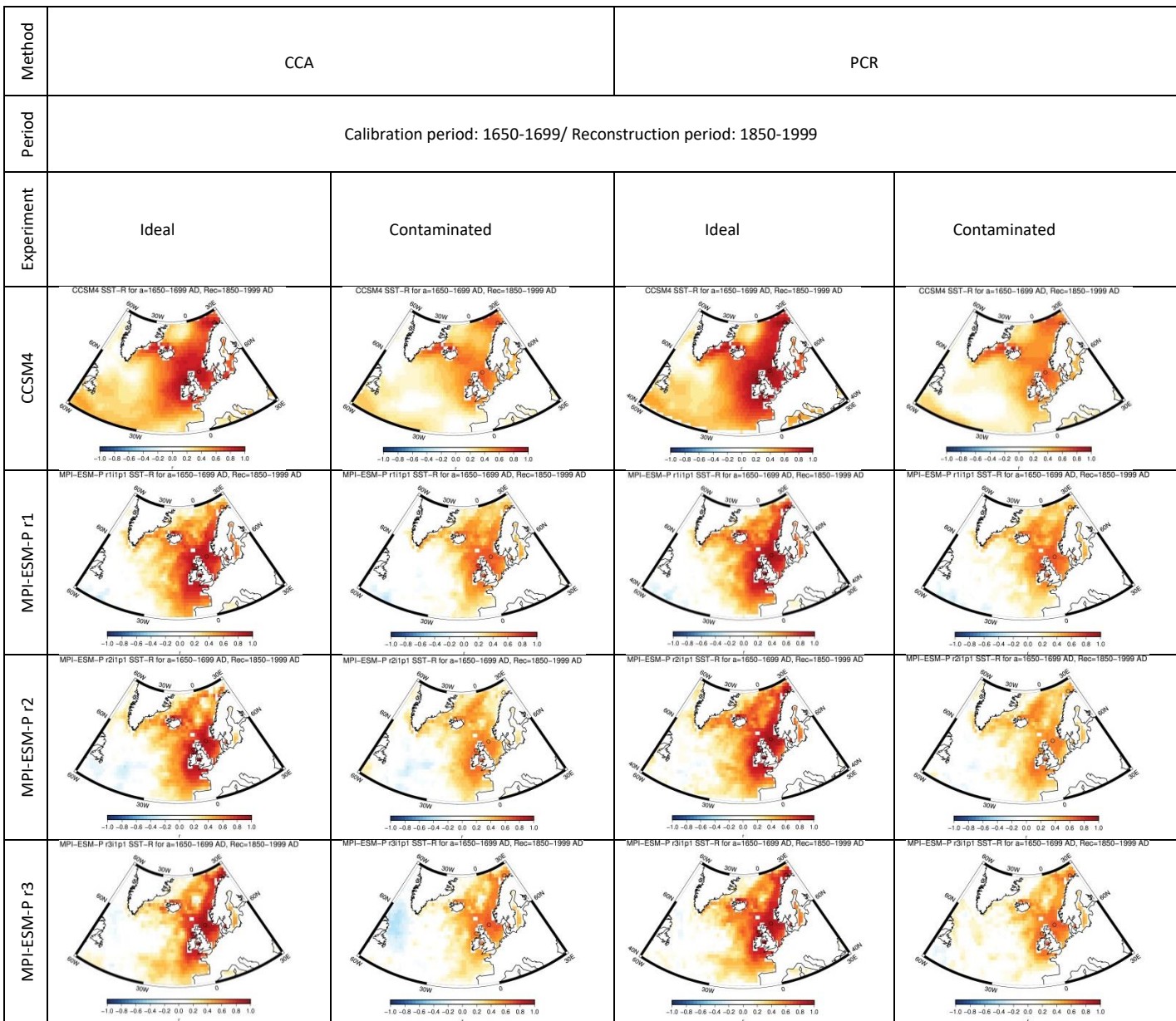

**Figure 2S: As in Fig. 1S, but the results are given for the LIA calibration period (1650—1699 AD).**



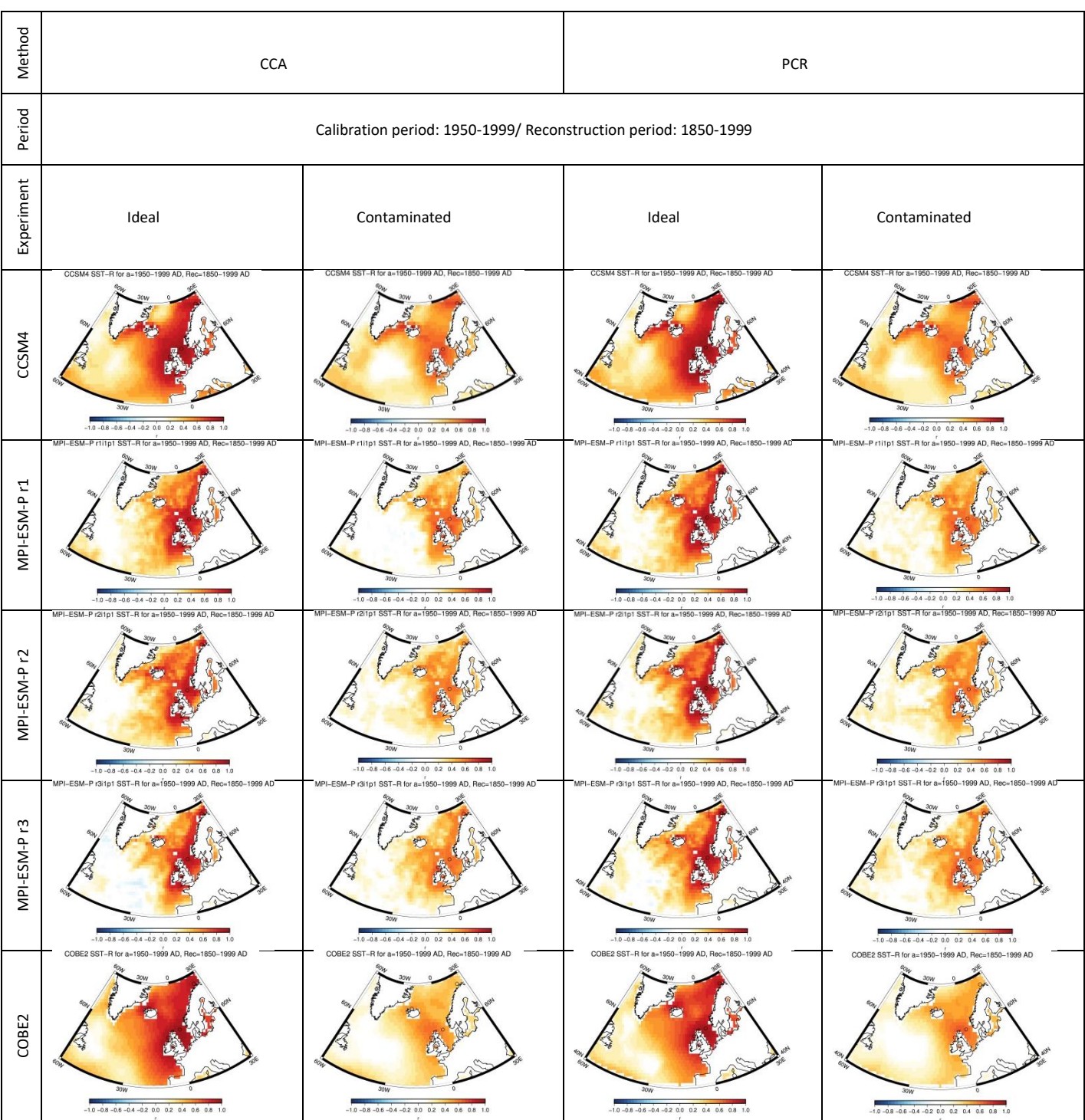

Figure 3S: As in Fig. 1S, but the results are given for the recent calibration period (1950—1999 AD). The last row contains the pseudo-proxy results of the COBE2 reanalysis data.

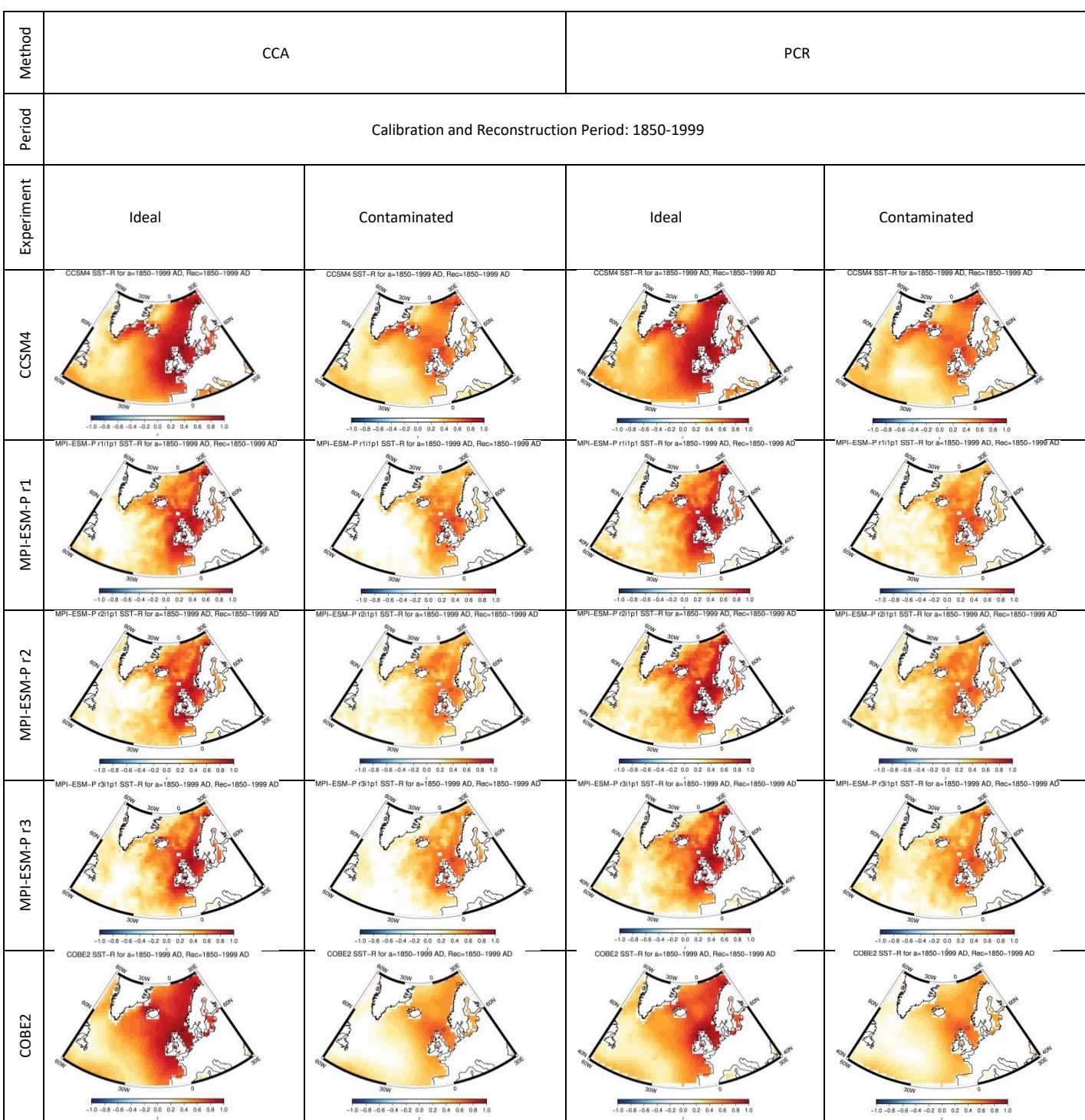

**Figure 4S: As in Fig. 3S, but the results are given for the industrial calibration period (1850—1999 AD).**



| Method | CCA | | PCR | |
|---|---|---|---|---|
| Period | Calibration and Reconstruction Period: 850-1849 | | | |
| Experiment | Ideal | Contaminated | Ideal | Contaminated |

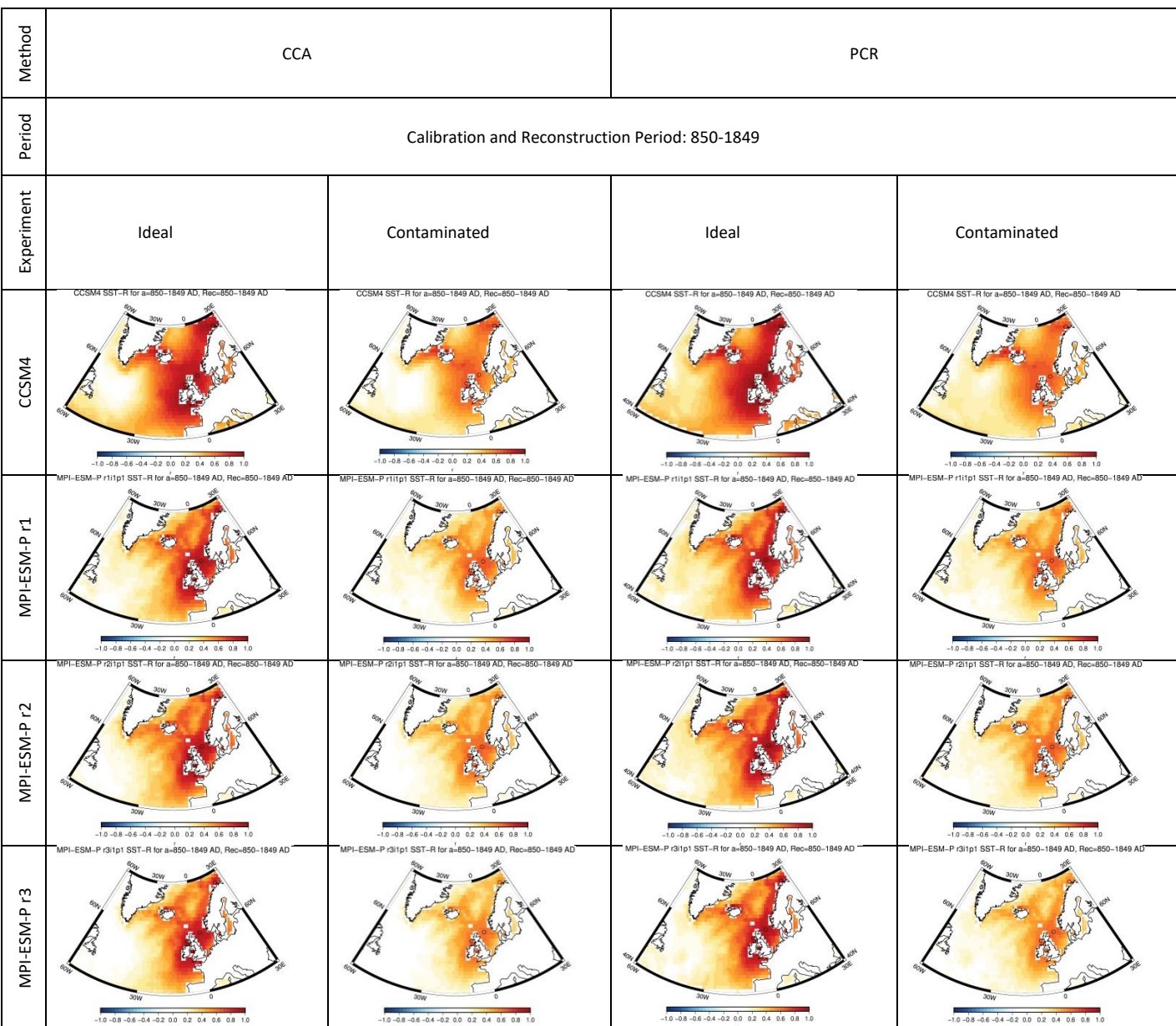

**Figure 5S: As in Fig. 1S, but the results are given for the pre-industrial calibration period (850—1849 AD).**




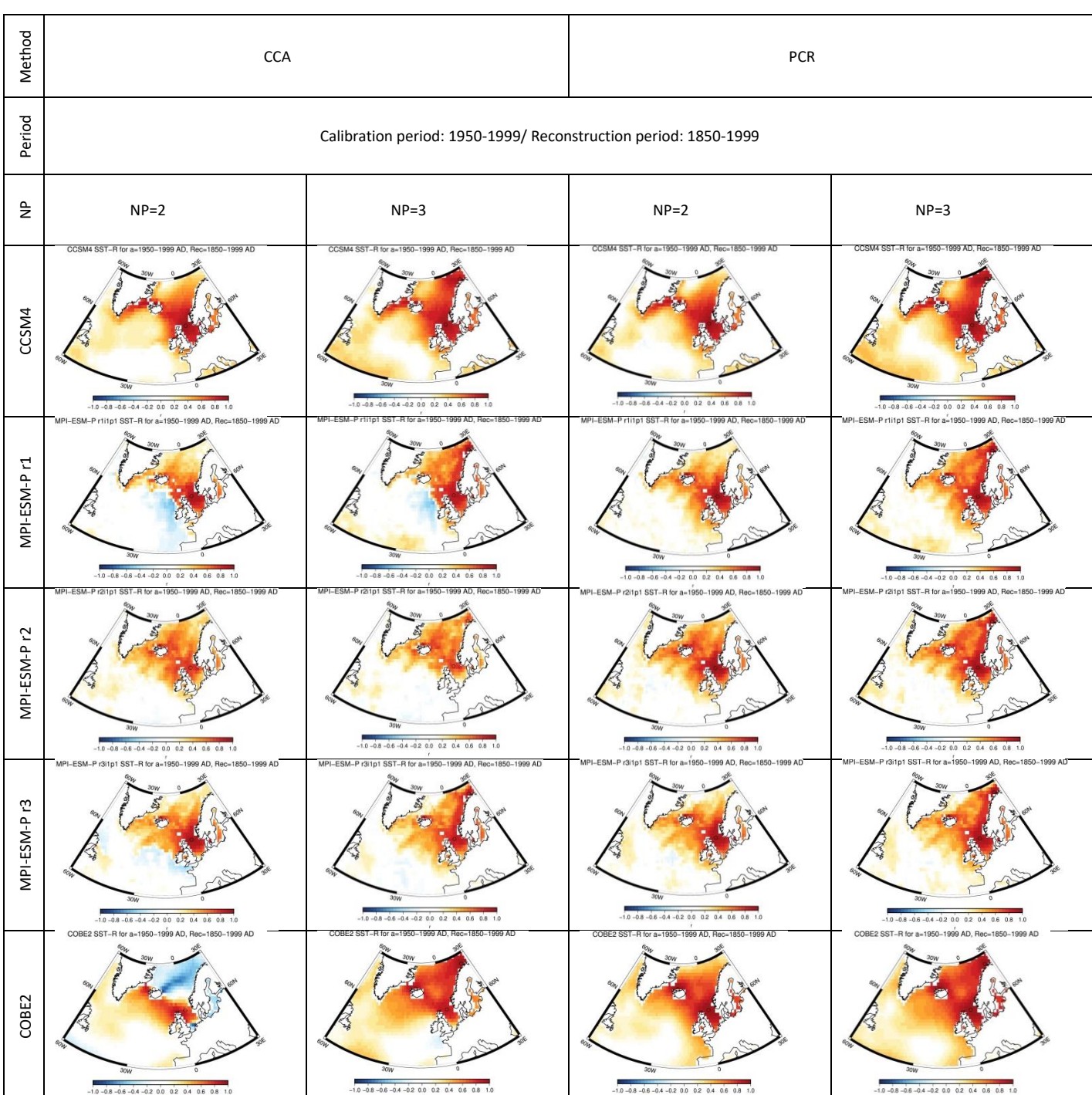

**Figure 6S: Correlation coefficient between the reconstructed and the original SST-anomaly evolution of the NA field during the industrial era, when the number of proxy locations used is NP=2 (1st and 3rd column) and NP=3 (2$^n$ and 4$^{th}$ column). The results are given for the recent calibration period (1950—1999 AD) and for the two different reconstruction methods (1$^{st}$ and 2$^{nd}$ column CCA, 3$^{rd}$ and 4$^{th}$ column PCR) for the model CCSM4 (1$^{st}$ row), three realizations of the MPI-ESM-P model (2$^{nd}$, 3$^{rd}$ and 4$^{th}$ row) and COBE2 reanalysis data (5$^{th}$ row).**

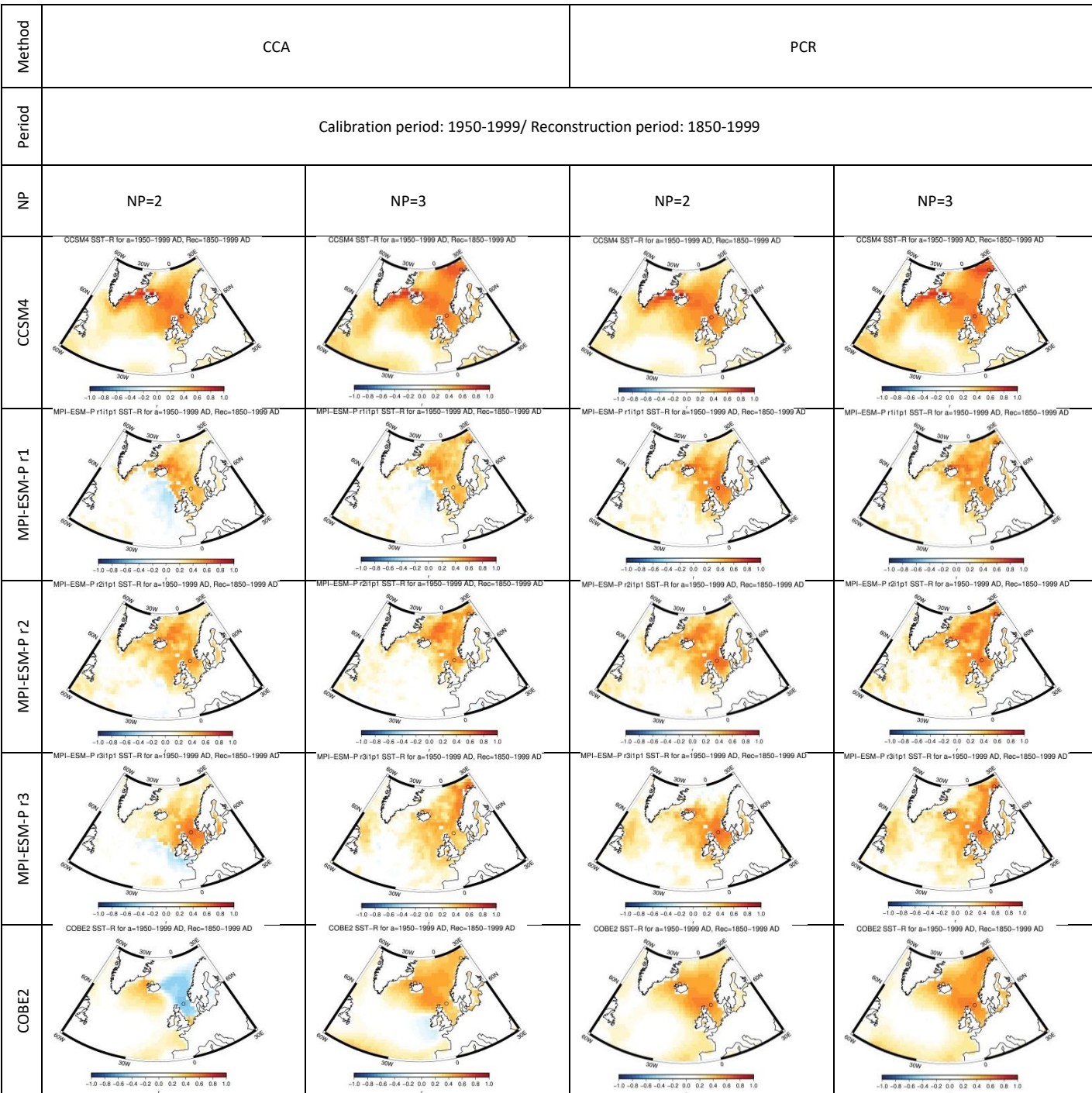


**Figure 7S: As in Fig. 6S, but the results are given for the noise contaminated pseudo-proxy experiment.**



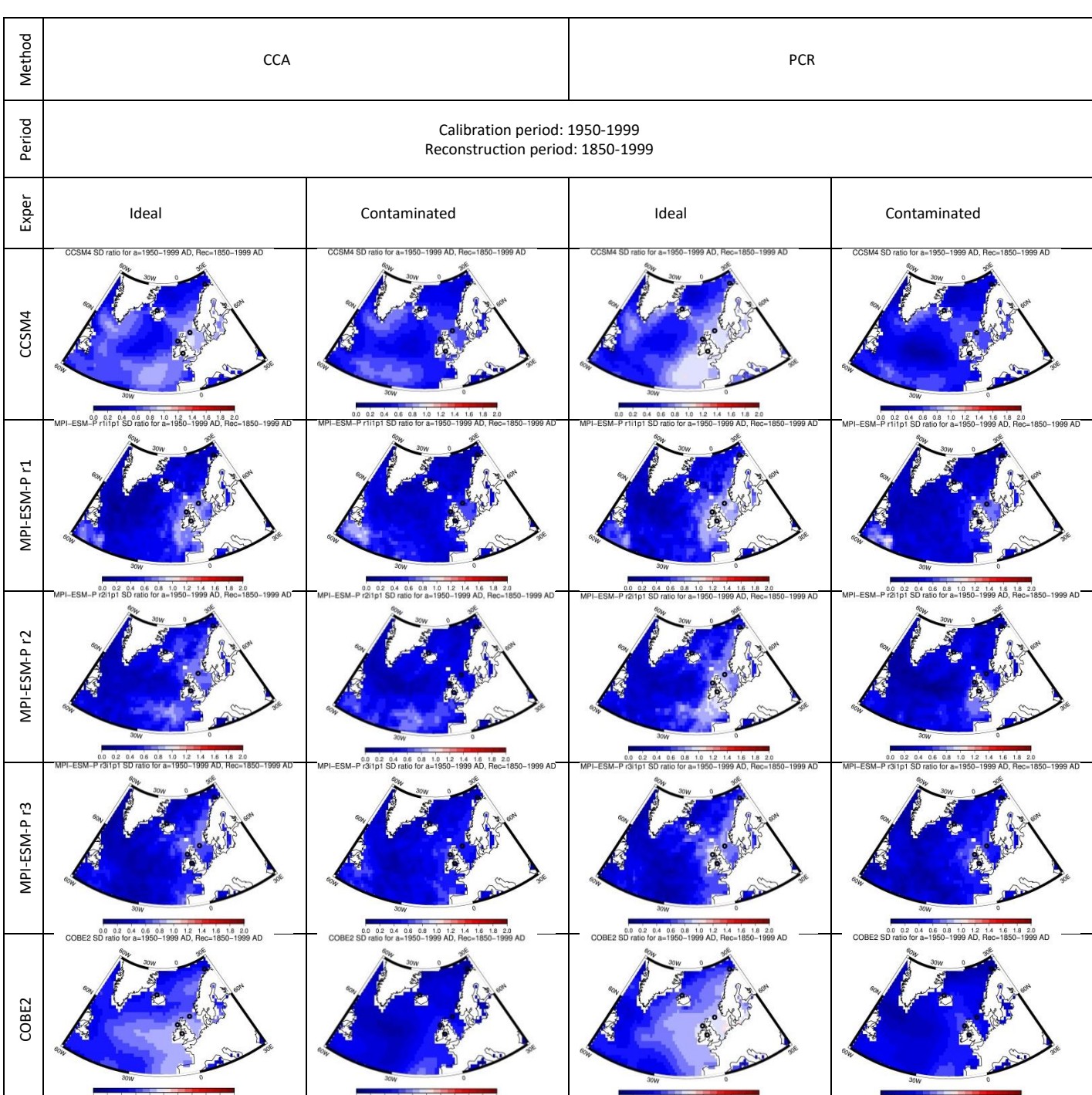

**Figure 8S: SD ratio between the reconstructed and the original SST-anomaly evolution of the NA field during the industrial period, for the ideal (column 1 and 3) and the noise contaminated (column 2 and 4) pseudo-proxy experiment. The results are given for the recent calibration and for the two different reconstruction methods (1st and 2nd column CCA, 3rd and 4th column PCR) for the models CCSM4 (1st row) and three realizations of the MPI-ESM-P model (2nd, 3rd and 4th row), as well as the COBE2 data (5th row).**



| Method | CCA | | PCR | |
|---|---|---|---|---|
| Period | Calibration period: 1650-1699 Reconstruction period: 1850-1999 | | | |
| Exper | Ideal | Contaminated | Ideal | Contaminated |
| CCSM4 | | | | |
| MPI-ESM-P r1 | | | | |
| MPI-ESM-P r2 | | | | |
| MPI-ESM-P r3 | | | | |

**Figure 9S: As in Fig. 8S, but for the LIA calibration period and for the model CCSM4 (1[st] row) and three realizations of the MPI-ESM-P model (2[nd], 3[rd] and 4[th] row).**




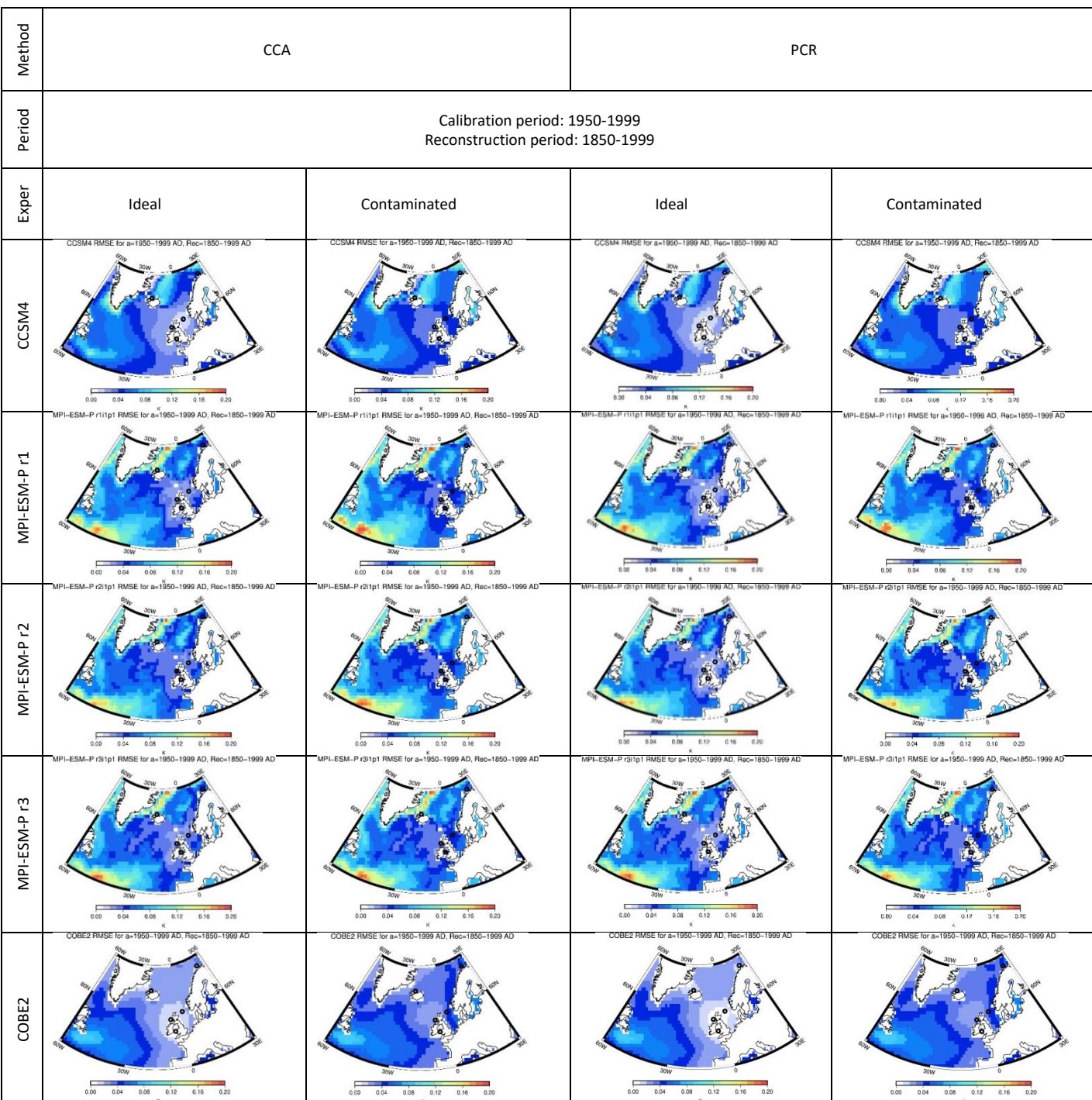

**Figure 10S: RMSE of the reconstructed SST-anomalies of the NA field during the industrial period, for the ideal (column 1 and 3) and the noise contaminated (column 2 and 4) pseudo-proxy experiment. The results are given for the recent calibration and for the two different reconstruction methods (1st and 2nd column CCA, 3rd and 4th column PCR) for the models CCSM4 (1st row) and three realizations of the MPI-ESM-P model (2nd, 3rd and 4th row), as well as the COBE2 data (5th row).**

| Method | CCA | | PCR | |
|---|---|---|---|---|
| Period | Calibration period: 1650-1699 Reconstruction period: 1850-1999 | | | |
| Exper | Ideal | Contaminated | Ideal | Contaminated |
| CCSM4 | | | | |
| MPI-ESM-P r1 | | | | |
| MPI-ESM-P r2 | | | | |
| MPI-ESM-P r3 | | | | |

**Figure 11S: As in Fig. 10S, but for the LIA calibration period and for the model CCSM4 (1st row) and three realizations of the MPI-ESM-P model (2nd, 3rd and 4th row).**




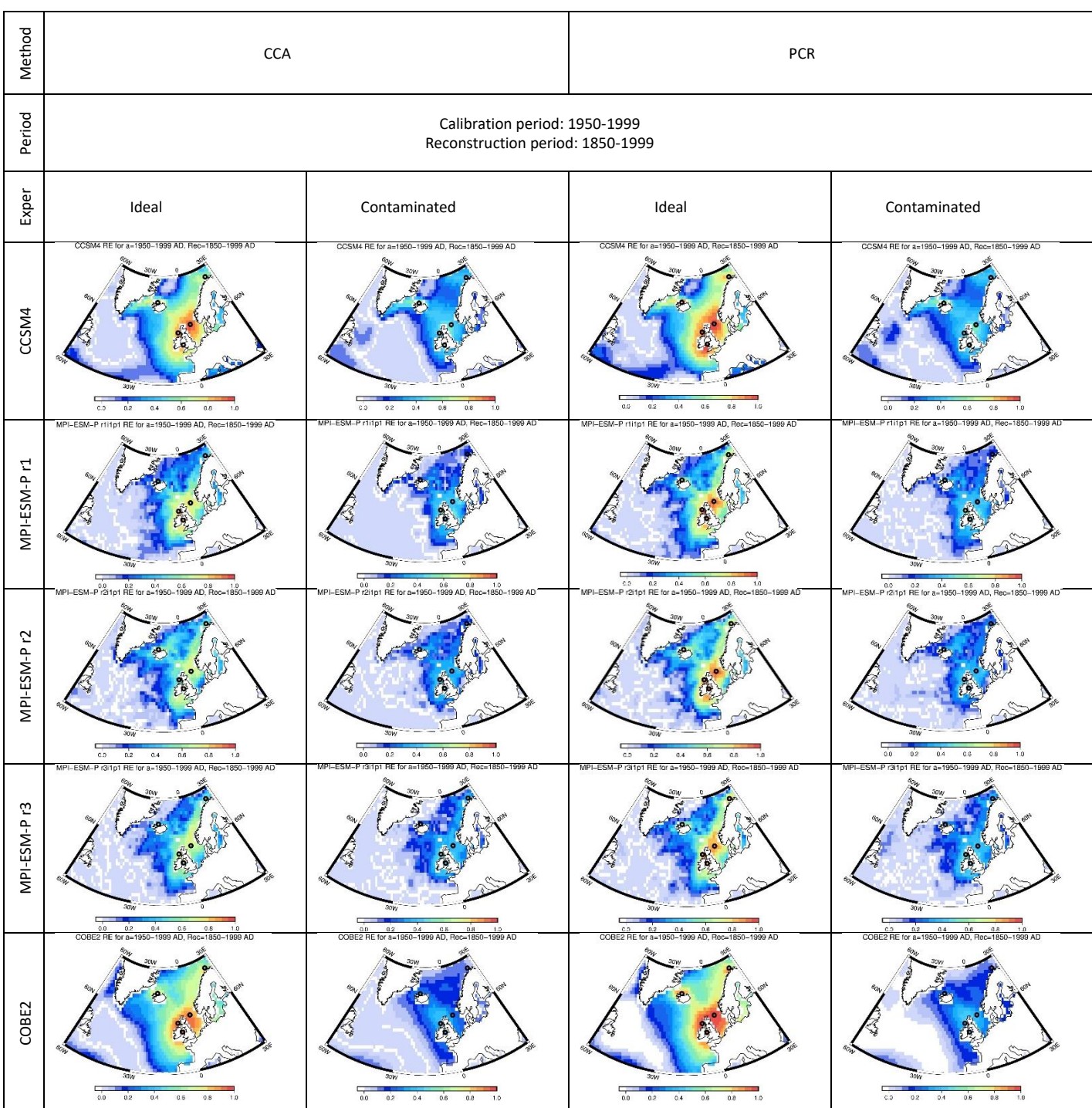

Figure 12S: RE of the reconstructed SST-anomalies of the NA field during the industrial period, for the ideal (column 1 and 3) and the noise contaminated (column 2 and 4) pseudo-proxy experiment. The results are given for the recent calibration and for the two different reconstruction methods (1st and 2nd column CCA, 3rd and 4th column PCR) for the models CCSM4 (1st row) and three realizations of the MPI-ESM-P model (2nd, 3rd and 4th row), as well as the COBE2 data (5th row).

| Method | CCA | | PCR | |
|---|---|---|---|---|
| Period | Calibration period: 1650-1699 Reconstruction period: 1850-1999 | | | |
| Exper | Ideal | Contaminated | Ideal | Contaminated |
| CCSM4 | | | | |
| MPI-ESM-P r1 | | | | |
| MPI-ESM-P r2 | | | | |
| MPI-ESM-P r3 | | | | |

**Figure 13S: As in Fig. 12S, but for the LIA calibration period and for the model CCSM4 (1st row) and three realizations of the MPI-ESM-P model (2nd, 3rd and 4th row).**




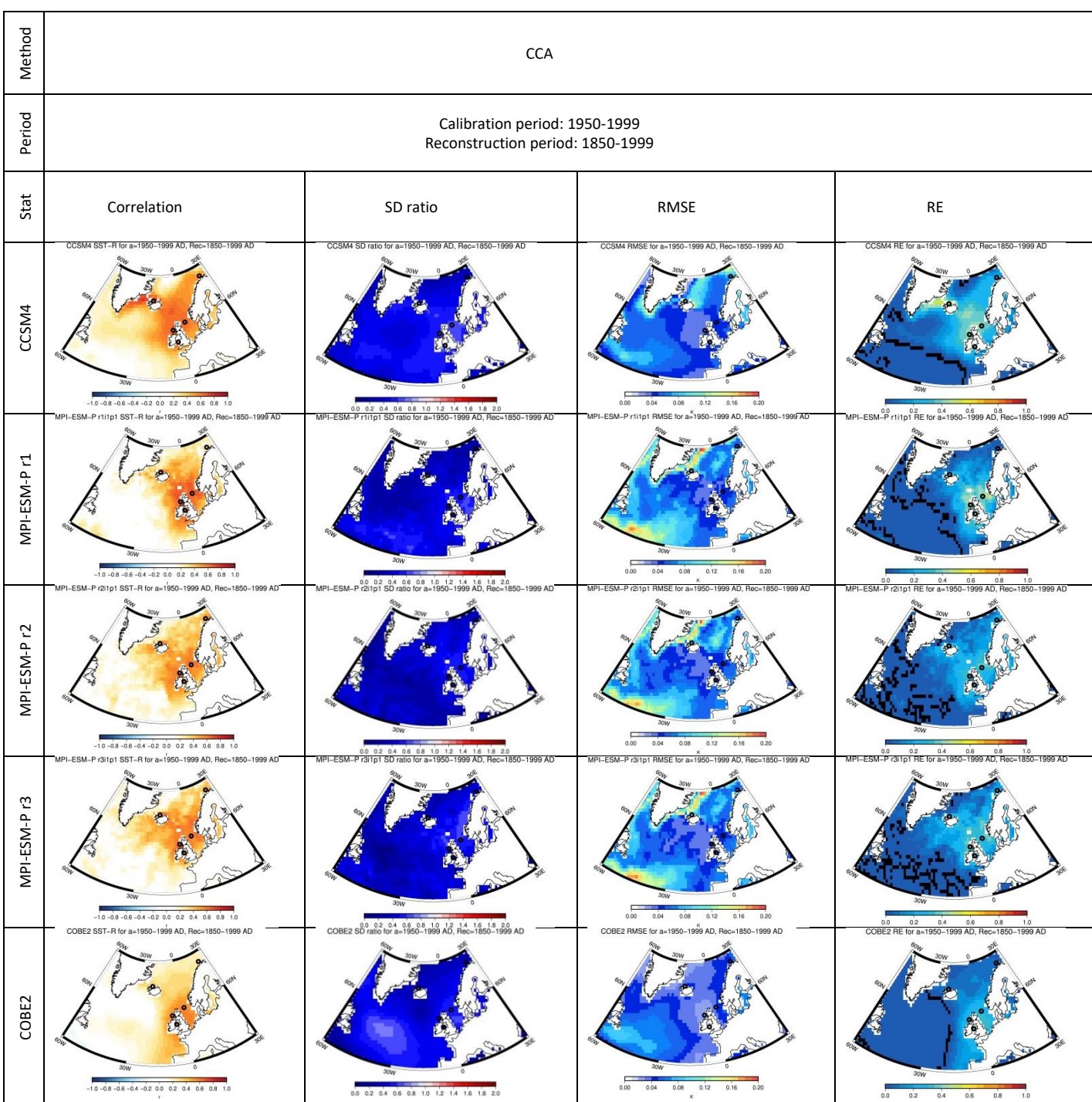

Figure 14S: The correlation and SD ratio between the reconstructed and the original SST-anomaly evolution of the NA field during the industrial period are given in columns 1 and 2, respectively. The RMSE and RE of the reconstructed SST anomalies are shown in columns 3 and 4 respectively. The results are calculated using CCA, regard a second noise realization of the pseudo-proxy experiment and are given for the recent calibration period and for the models CCSM4 (1st row) and three realizations of the MPI-ESM-P model (2nd, 3rd and 4th row), as well as the COBE2 data (5th row).

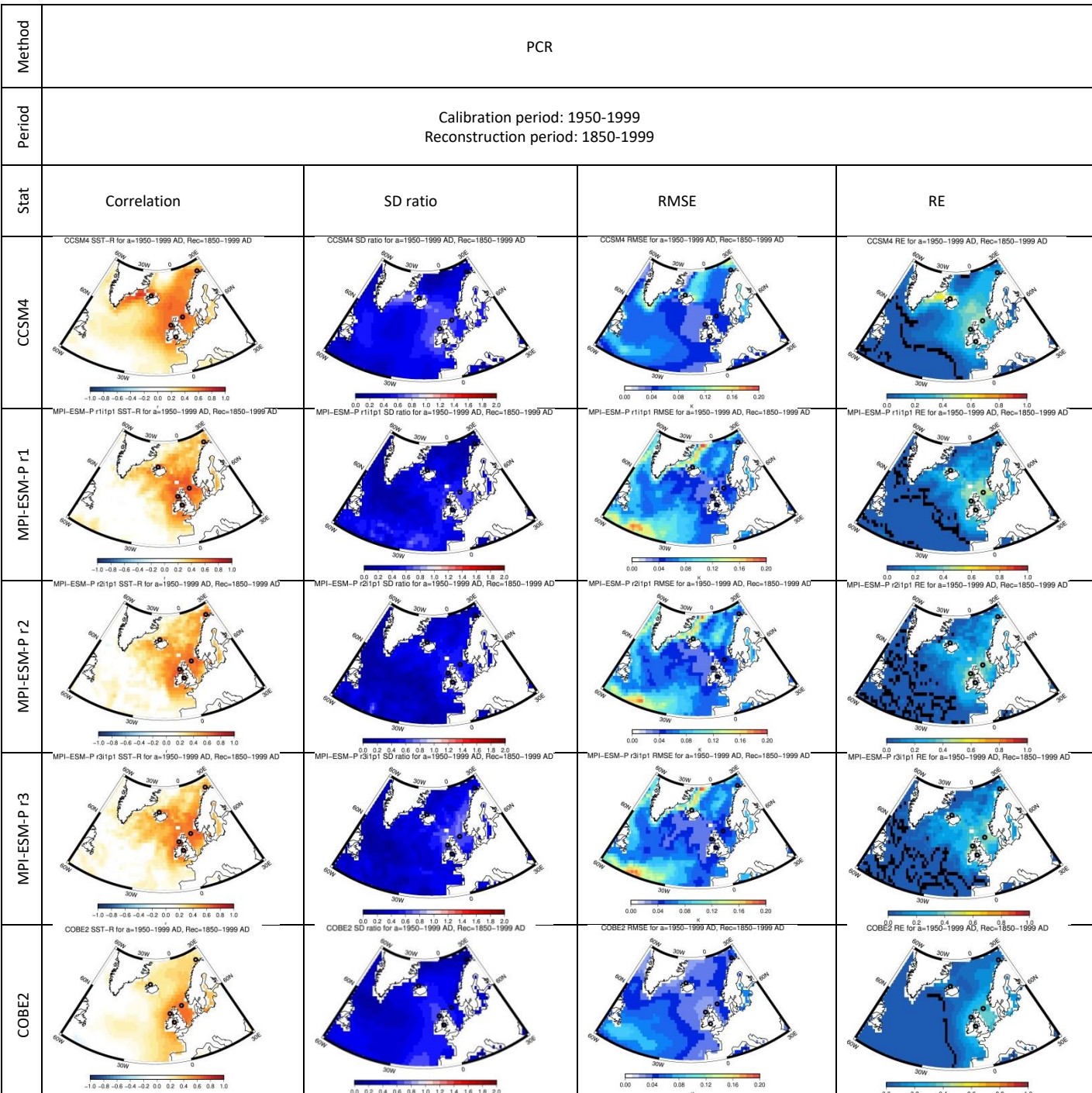

**Figure 15S: As in Figure 14S, but the results are calculated using PCR.**



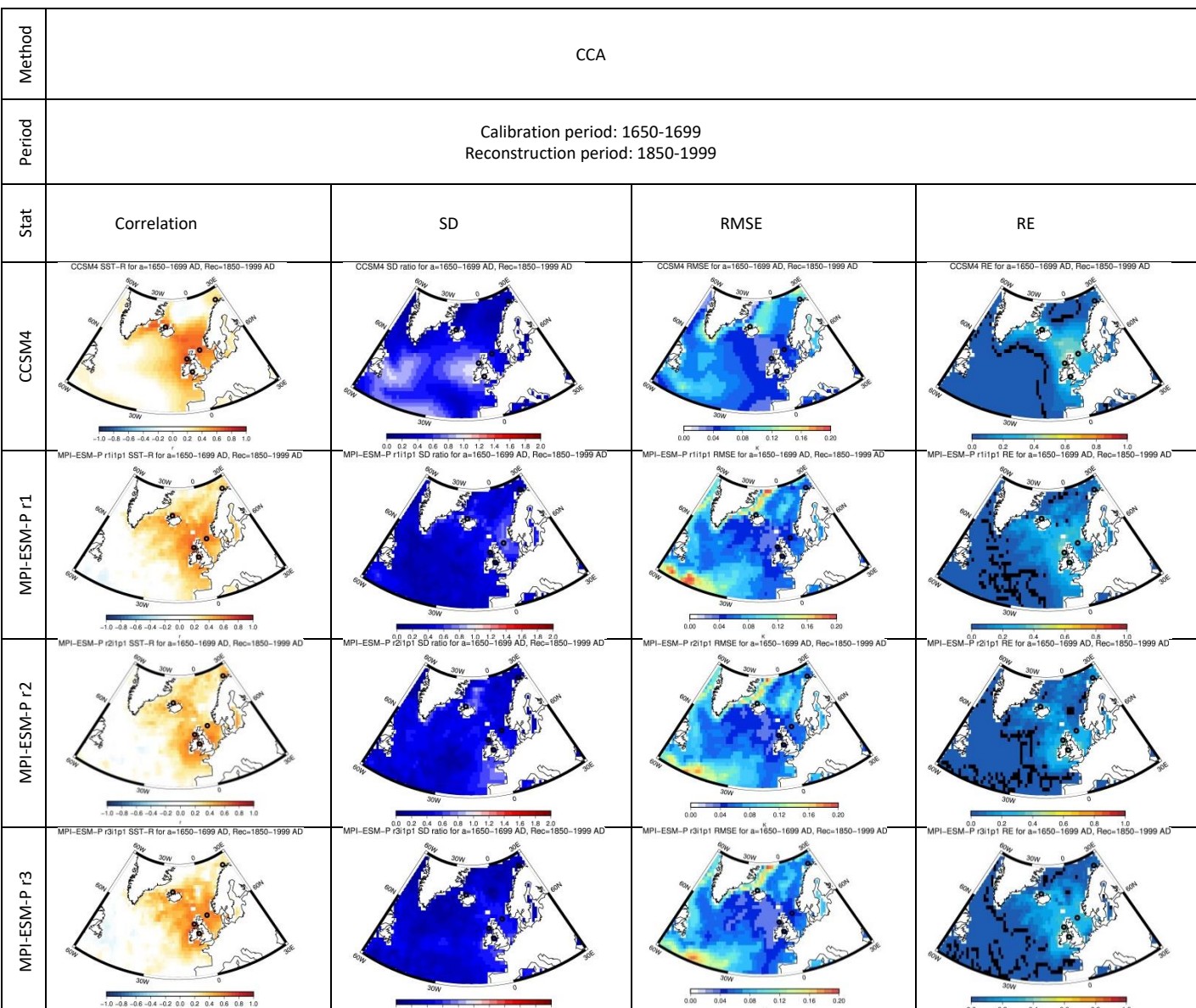

**Figure 16S:** The correlation and SD ratio between the reconstructed and the original SST-anomaly evolution of the NA field during the industrial period are given in columns 1 and 2, respectively. The RMSE and RE of the reconstructed SST anomalies are shown in columns 3 and 4 respectively. The results are calculated using CCA, regard a second noise realization of the pseudo-proxy experiment and are given for the LIA calibration period and for the models CCSM4 (1st row) and three realizations of the MPI-ESM-P model (2nd, 3rd and 4th row), as well as the COBE2 data (5th row).

| Method | PCR | | | |
|---|---|---|---|---|
| Period | Calibration period: 1650-1699<br>Reconstruction period: 1850-1999 | | | |
| Stat | Correlation | SD | RMSE | RE |
| CCSM4 | | | | |
| MPI-ESM-P r1 | | | | |
| MPI-ESM-P r2 | | | | |
| MPI-ESM-P r3 | | | | |

**Figure 17S: As in Fig. 16S, but the results are calculated using PCR.**





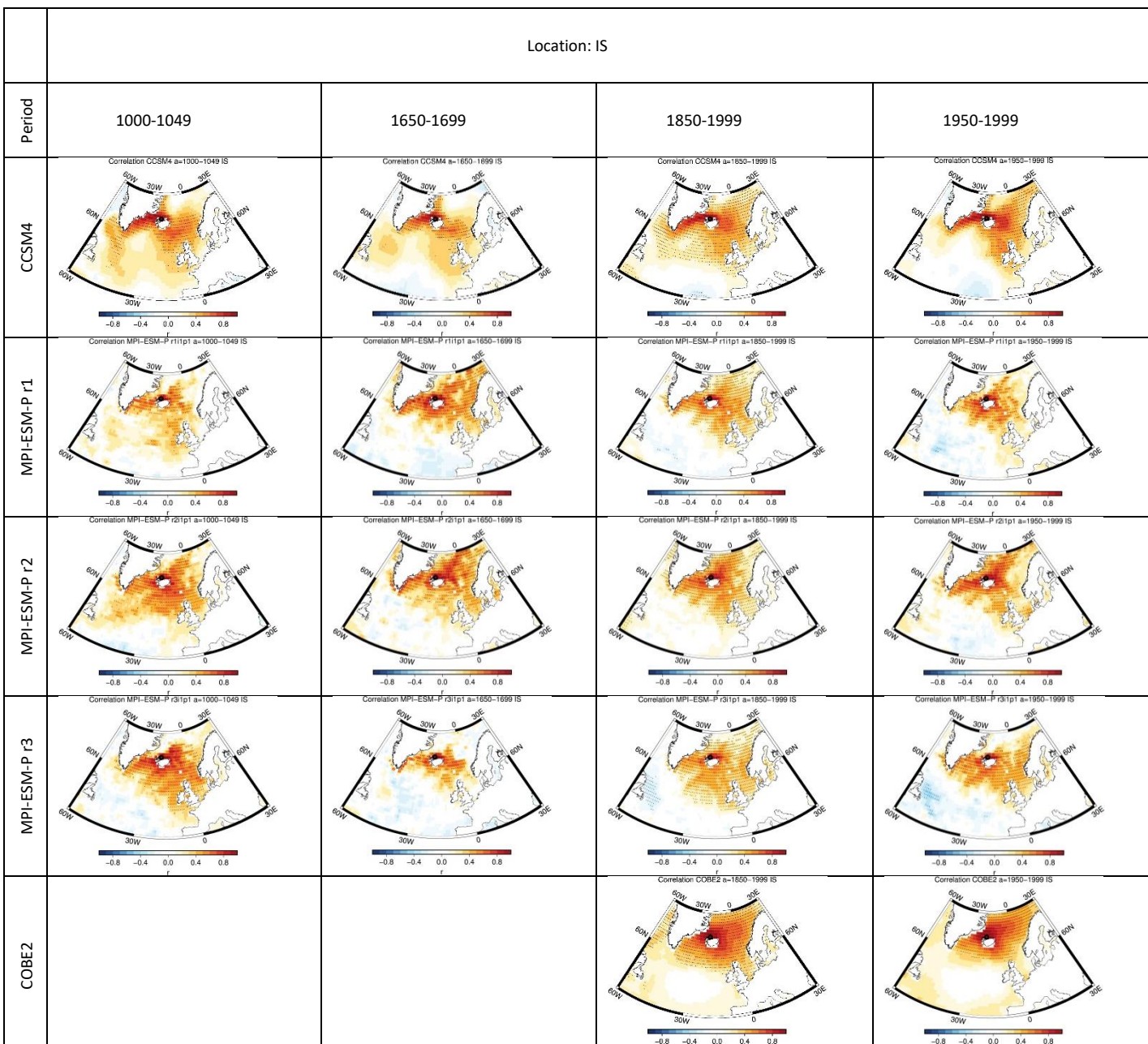

**Figure 18S:** One point correlation maps between the SSTs co-located to the IS site and the NA basin, for the MCA period (1ˢᵗ column), the LIA period (2ⁿᵈ column), the industrial period (3ʳᵈ column) and the recent period (4ᵗʰ column). Hatched areas indicate statistical significance on the 99% level.




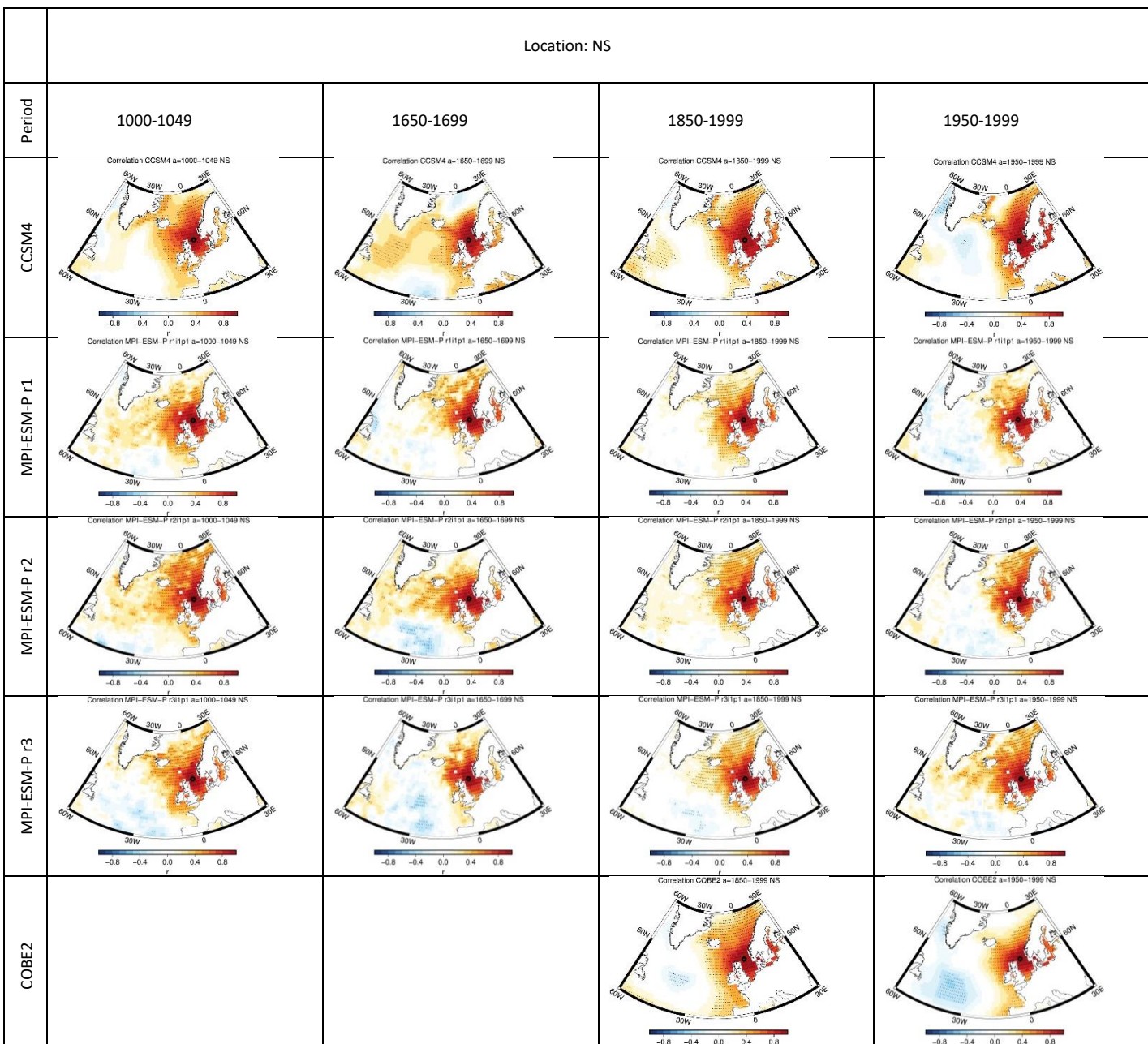

**Figure 19S: As in Figure 18S, but for the NS site. Hatched areas indicate statistical significance on the 99% level.**




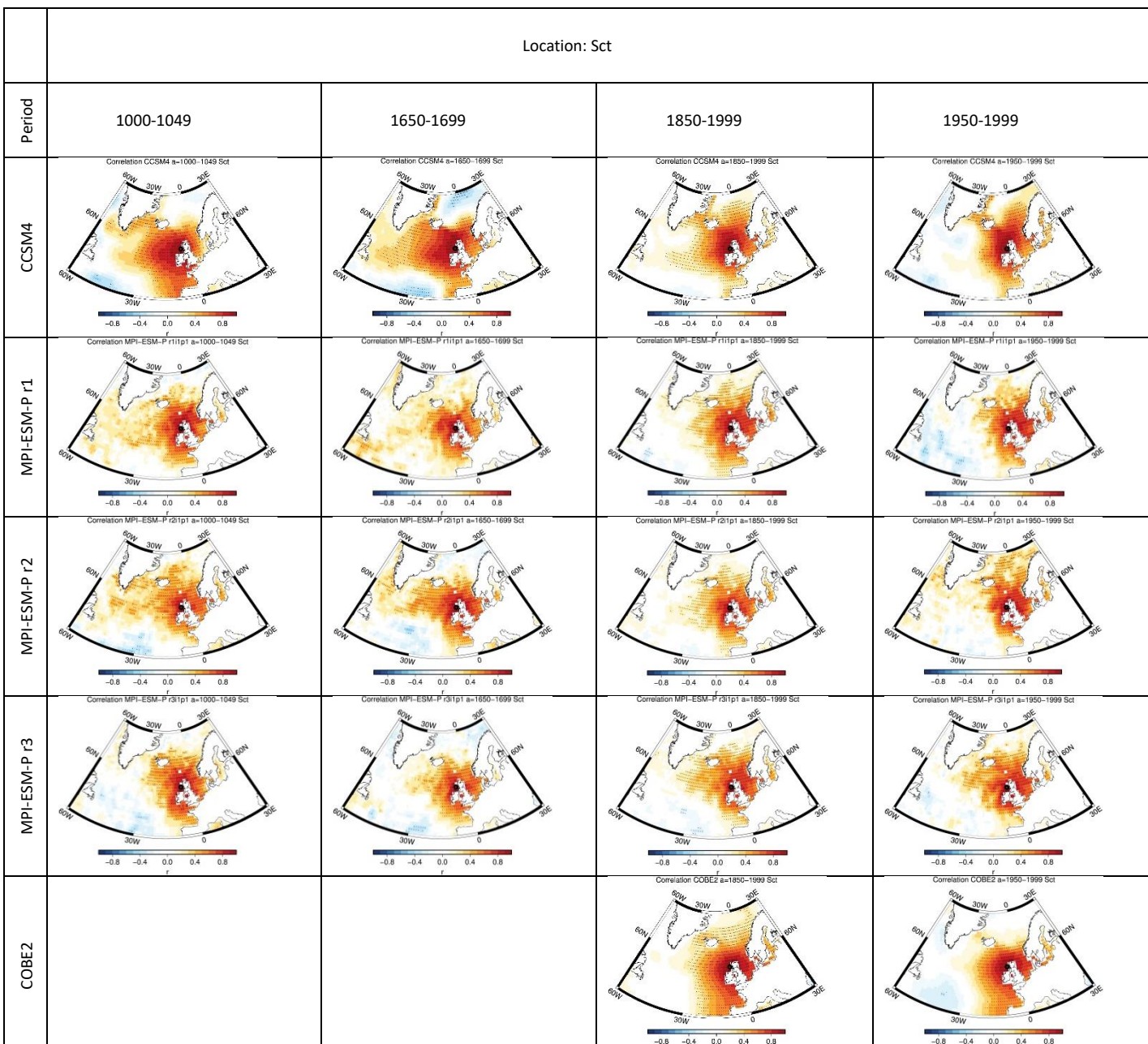

**Figure 20S: As in Figure 18S, but for the Sct site. Hatched areas indicate statistical significance on the 99% level.**




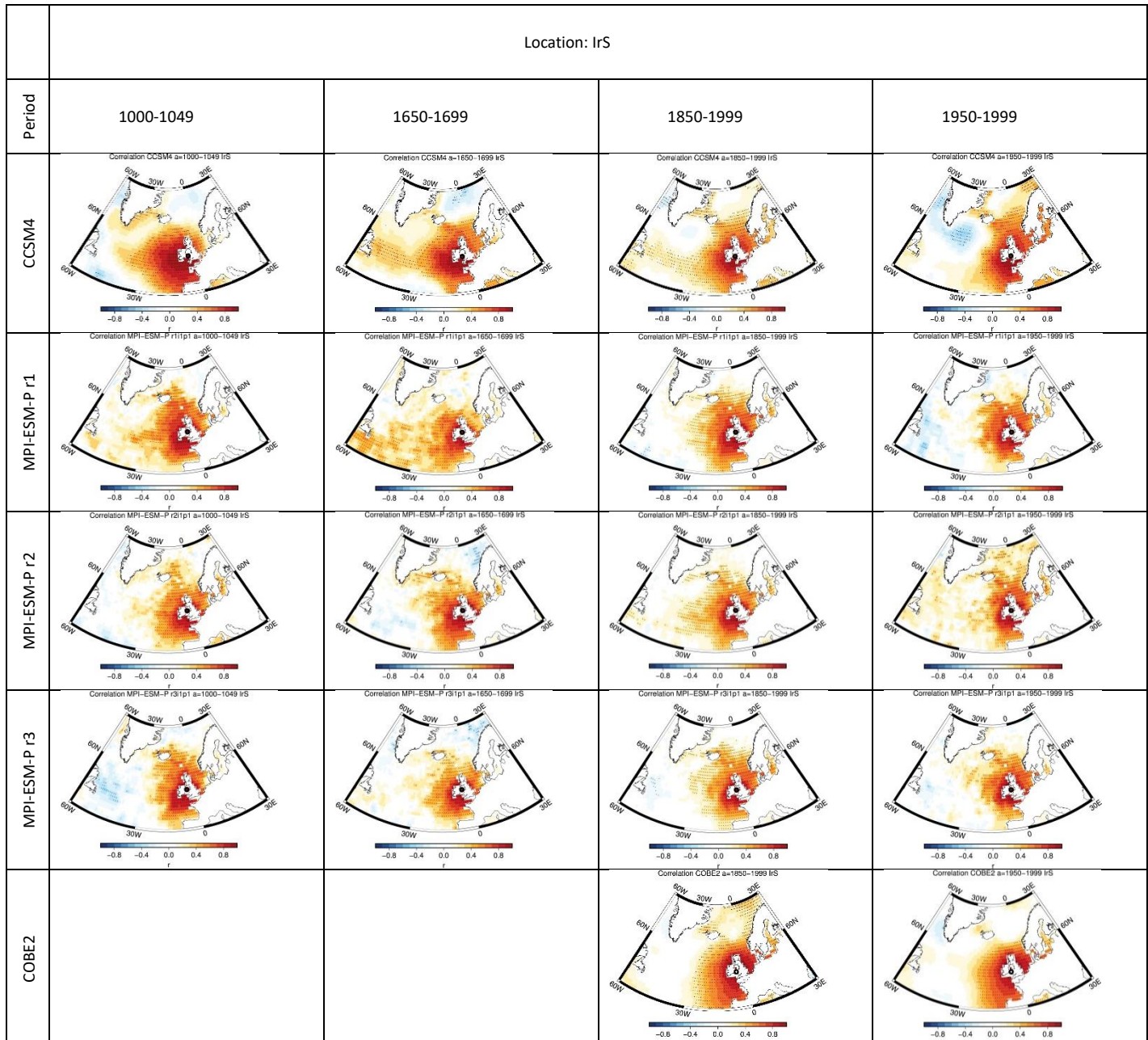

Figure 21S: As in Figure 18S, but for the IrS site. Hatched areas indicate statistical significance on the 99% level.

| | Location: InI | | | |
|---|---|---|---|---|
| Period | 1000-1049 | 1650-1699 | 1850-1999 | 1950-1999 |

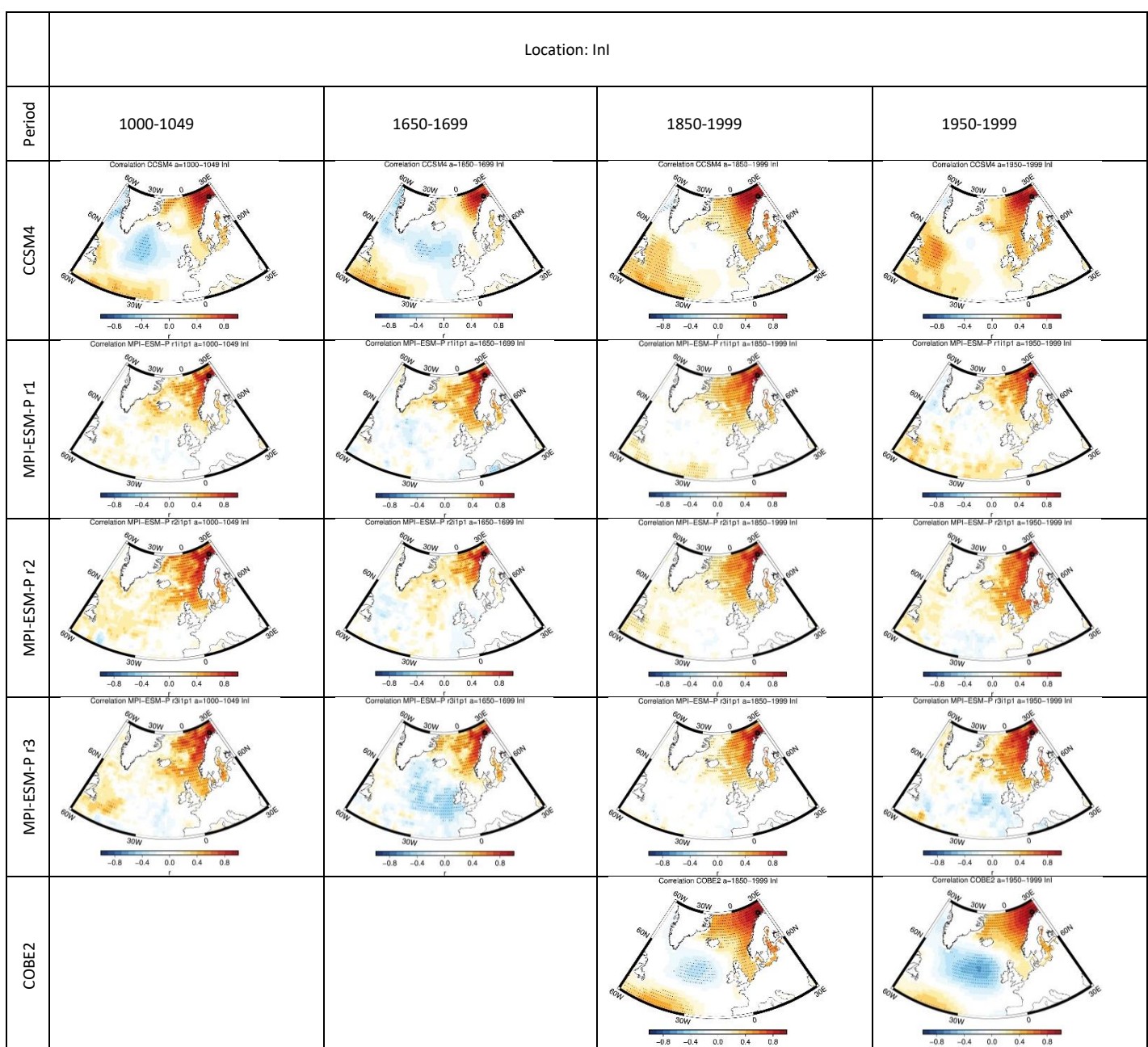

**Figure 22S: As in Figure 18S, but for the InI site. Hatched areas indicate statistical significance on the 99% level.**