# Peer review of "Pseudo-proxy evaluation of Climate Field Reconstruction methods of North Atlantic climate based on an annually resolved marine proxy network"

_Climate of the Past, 2017_

## Referee Comment (RC1) · Anonymous Referee #1 · 12 May 2017

The manuscript "Pseudo-proxy evaluation of Climate Field Reconstruction methods of North Atlantic climate based on an annually resolved marine proxy network" represents a good pseudo-proxy experiment on the efficacy of using proxy records of the bi-valve mollusk Arctica islandica to reconstruct interannual variations in SSTs in the northeast Atlantic Ocean. Based on the experiments reported, especially from the more realistic noise-degraded versions, the use of A. islandica is indicated as suitable for this purpose, specifically in the region surrounding the British Isles and the open ocean region between there, Iceland, and Norway.

I believe this paper will be suitable for publication in Climate of the Past once a number of relatively minor revisions are made. I've noted these in the attached supplement file, which contains the originally-submitted manuscript with edits for usage along with comments noted using the Adobe Comment facilities for PDF documents.

The most important comment I have from a substantive standpoint relates to the definition of the Z noise variable in Equation 6. This nature of this random variable is critical to how the noise degradation for the pseudo-proxy experiments is defined. It needs to be described for the readers, so they can make judgements for themselves about the experimental results.

Also, in a number of places the references to existing literature need to be updated or extended, as noted in the comment boxes.

Overall, I think this is a good paper, and with the changes noted, will make a good contribution to the literature and provide a reasonable basis on which to attempt a real-world reconstruction of SST's in the region mentioned using A. islandica proxy data.

Please also note the supplement to this comment:
http://www.clim-past-discuss.net/cp-2017-61/cp-2017-61-RC1-supplement.pdf

**Supplement:**

[revised manuscript text omitted]

---

## Referee Comment (RC2) · Anonymous Referee #2 · 17 May 2017

The paper investigates the skill of two different field reconstruction methods when applied to a handful of proxy locations in the North Atlantic ocean. The authors generate pseudo-proxies based on SSTs from two different models and an observed data set. They conclude that proxies at these locations have skill in reconstructing the temperature field in the eastern North Atlantic.

Several previous papers have studied the skills of field reconstruction methods. The present paper differs from these by focusing on a smaller region and on a very small set of proxy locations (5 or less). I find the paper somewhat limited in scope and I can only recommend that the paper is accepted if the authors expand their analysis to include more and deeper explanations of the results.

Major comments:

1) The pseudo-proxy experiments seem to be done for only one realization of the AR1 noise even though several models are used. Many of the conclusions – which are based on rather small differences in the correlation maps – could be different if another noise realization was used as shown in, e.g., Christiansen et al. 2009 (doi: 10.1175/2008JCLI2301.1). Preferably an ensemble of realizations should be used. Alternatively different realizations should be performed and the (possible) differences discussed.

2) Both the considered methods depend on EOF analysis and section 4.1 discusses the stationarity of the calibration coefficients. However, the paper does not include any comparison of the EOFs or the spatial correlation structure. I would advise the authors to compare the EOFs from the different data-sets and the different periods. The stationarity of the teleconnections could also be investigated by looking at the map of correlations between a grid-point with proxy-data and all other points

3) In the conclusions and the abstract it is mentioned that the marine network can produce skillful spatial reconstructions for the eastern NA basin. But even in this area there seems to be a massive underestimation of the amplitude. I think this underestimation in general could be described more in both the text and in the abstract.

4) I guess the reconstructions are best in the areas close to the proxies. But I don't think this is discussed much in the paper. The position of the proxies could be indicated on the maps.

5) It would be nice to see a plot of the time-series of the 5 real-world proxies. This would also allow the reader to judge if the AR1 process used for the pseudo-proxies is sound. By the way, I am surprised that the authors did not show a real-world reconstruction based on the 5 Arctica islandica.

6) There has been a discussion in the literature of the reconstruction methods ability to

get the amplitude right. Methods with temperature as the dependent variable, as those used here, are prone to underestimate the variability (Christiansen and Ljungqvist 2017, doi: 10.1002/2016RG000521 and references therein). The results in this paper seem to get an underestimation of the variability even in the case with noise-free proxies. The reason for the underestimation of the variability should be investigated and discussed.

Minor comments:

line 35: I think the abbreviations such as NA and PPE should also be defined in the text and not just in the abstract.

line 54: Perhaps another word than "aggravating" should be used here.

line 92: grid point -> gridded.

line 111 and 30: It is confusing that "reconstructions" are used for different things.

line 119: Why 1999 and not 2005?

line 142: A degree sign is missing.

Section 2.2.1: It would be better if single-letter symbols would be used in the formulas instead of Proxy, EOF, CC etc.

line 193: The sentence beginning with "The key .." does not make much sense to me.

l205: The pseudo-proxy review Smerdon 2001 (doi: 10.1002/wcc.149) should be cited somewhere.

line 200: With only five (or less in section 3.3) proxies it does not seem to make any sense to make a EOF transformation of the proxies and keep all five modes. This step is usually done to reduce the number of degrees of freedom which is not necessary here. This should be discussed.

line 223: What are the correlation of the other 4 proxies? It could be noted that values

around 0.4 are quite characteristic for many other proxies.

---

## Referee Comment (RC3) · Anonymous Referee #3 · 22 May 2017

General comments:

Pyrina et al. test the ability of two climate field reconstruction methods (PCR and CCA) to reconstruct sea surface temperatures in the North Atlantic based on pseudoproxy experiments (PPEs) that replicate the spatial locations and noise characteristics of a small Arctica islandica proxy network. They show that, within the context of these PPEs, both PCR and CCA can produce reasonable skillful reconstructions of sea surface temperatures in parts of this region, but with differences depending on which model/reanalysis dataset is used as the target field. Overall, I think the paper presents some interesting results and is mostly well-written (though see "Technical Corrections" below). I do think the paper could benefit by being a little more explicit about

the research question(s), objective(s), key results, and (especially) the overall significance/contributions of the findings. As it stands, the broader significance of this work is a little unclear. I also have several specific concerns (listed in "Specific Comments") that I think the authors should address prior to publication.

Specific comments:

1) Why choose calibration periods that are earlier than the validation period? Given that every real-world climate reconstruction would be calibrated during the later period during which instrumental data is available, wouldn't it make more sense to calibrate the PPEs using the 1850-1999 period and then validate on the earlier time periods (Medieval, LIA, etc)? To me, that seems more intuitive and would still address the stationarity issue.

2) Is there any reason why these two particular CFR methods (PCR and CCA) were chosen, while other common CFR methods (e.g., RegEM-TTLS and RegEM-ridge) were excluded? I'm not necessarily suggesting that the authors need to redo the analyses with additional CFR methods, but I would at least like to see a little justification for why these methods were chosen while others were excluded.

3) I think it would be useful to include other complementary validation statistics, such as mean bias, coefficient of efficiency (CE), reduction of error (RE), and/or root mean squared error (RMSE). I'm not sure that only correlation and standard deviation ratio are enough for a robust assessment of model performance. A spatial assessment of mean bias and either CE or RMSE could add important information to this study.

4) How is it possible to have a standard deviation ratio greater than 1 for the CCA results? As I understand it, all parametric reconstruction methods will result in at least some variance loss unless the proxy is perfectly correlated with the target climate variable (McCarroll et al. 2015), which would only be the case at the particular grid cells with the noise-free pseudoproxies. Even in other studies that used noise-free pseudoproxies (e.g., Smerdon et al. 2010, 2011), at least some variance loss was observed.

[Figure]

I therefore don't see how the reconstructed SSTs could have grid cells with greater variance than the "observed" (or in this case, model-simulated) SSTs.

5) The authors state that they chose to retain the first 10 PCs in PCR and the first 5 EOFs in CCA. How sensitive are the results to this choice? How were these thresholds chosen? Why were different thresholds chosen for the two CFR methods? As it stands, these seem like arbitrary choices. Did the authors consider more objective criteria for determining these thresholds, such as the "estimated noise continuum" approach used by Mann et al. 2007 or an optimization approach similar to Smerdon et al. 2010?

6) I would like to see some more discussion about the stationarity of the PPEs. Specifically, there is an implicit assumption when creating the pseudoproxies that the proxy response to SST variability is stationary (since the pseudoproxies are just SST+noise). Is this necessarily a realistic assumption for real-world A. islandica, or is it possible that the response of this species to SST variation could be non-stationary (similar to the well-known "divergence problem" in high latitude tree-ring widths)?

Technical corrections:

-Line 32: "loosing" should be changed to "losing"

-Line 39: delete "target to"

-Lines 52-54: I would rephrase this sentence. It is not clear as is. I think "aggravating" is the wrong word choice.

-Line 65: "recent years" should be letter "e" not "c".

-Line 74: delete the comma after "PPEs"

-Lines 97-100: It might be nice to see the proxy locations in a figure instead of just listing the coordinates. I don't see this currently in the figures, and I think it would be helpful for interpretation of the results.

-Line 264: "extend" should be changed to "extent"

-Figure labels: these are labeled "PCA" but throughout most of the paper this CFR method is referred to as PCR. I would suggest changing the figure labels to PCR to match the term used in the main text (though there are some inconsistencies throughout the text regarding use of PCA vs. PCR).

References:

McCarroll, D., Young, G. H., & Loader, N. J. (2015). Measuring the skill of variance-scaled climate reconstructions and a test for the capture of extremes. The Holocene, 25(4), 618–626.

Mann, M. E., Rutherford, S., Wahl, E., & Ammann, C. (2007). Robustness of proxy-based climate field reconstruction methods. Journal of Geophysical Research, 112, D12109.

Smerdon, J. E., Kaplan, A., Zorita, E., González-Rouco, J. F., & Evans, M. N. (2011). Spatial performance of four climate field reconstruction methods targeting the Common Era. Geophysical Research Letters, 38, L11705.

Smerdon, J. E., Kaplan, A., Chang, D., & Evans, M. N. (2010). A Pseudoproxy Evaluation of the CCA and RegEM Methods for Reconstructing Climate Fields of the Last Millennium. Journal of Climate, 23(18), 4856–4880.

---

## Author Comment (AC1) · 21 Jun 2017

Dear editor,

Thank you very much for handling this manuscript. In the revision stage we will update and extend the citations of existing work including the recommended ones.

---

## Author Comment (AC3) · 21 Jun 2017

Answer by M. Pyrina et al. to: Anonymous Referee #2

We would like to thank the Referee #2 for the constructive comments. We agree that our focus area is smaller compared to other studies that evaluate the spatial skill of reconstruction methods, but even though the region is smaller it regards the marine environment. This is a region that lacks past information with high temporal resolution and the reconstruction methods are tested in this region for the first time in the context of an absolutely dated annually resolved marine proxy. Even though the aforementioned differences to other studies are stated in the introduction we will reformulate the

introduction in order to account for the broader significance of the work in a more clear way. In some cases we will include more results and further discussion, as suggested by the reviewer, and provide more explanations of our findings.

Major comments:

1) The pseudo-proxy experiments seem to be done for only one realization of the AR1 noise even though several models are used. Many of the conclusions – which are based on rather small differences in the correlation maps – could be different if another noise realization was used as shown in, e.g., Christiansen et al. 2009 (doi: 10.1175/2008JCLI2301.1). Preferably an ensemble of realizations should be used. Alternatively different realizations should be performed and the (possible) differences discussed.

Even though we think that an additional noise realization of the AR1 model will not produce results that will change the main conclusions of our work we will conduct our experiments with additional noise realizations of the AR1 model and include the results in the text. In this context we will choose a range of the parameter of the upper AR1 model that can be realistically expected from the real noise level contained in the proxy data.

2) Both the considered methods depend on EOF analysis and section 4.1 discusses the stationarity of the calibration coefficients. However, the paper does not include any comparison of the EOFs or the spatial correlation structure. I would advise the authors to compare the EOFs from the different data-sets and the different periods. The stationarity of the teleconnections could also be investigated by looking at the map of correlations between a grid-point with proxy-data and all other points.

Comparing the EOFs of the different periods will not address the stationarity of the calibration coefficients, as the reconstruction was performed in every case by using the leading EOFs calculated in the calibration period. So, even though the individual EOFs might be different between different periods, the leading EOFs from i.e. the period

1950-1999 will roughly capture the same co-variability as the leading EOFs of another period and therefore no conclusions about the stationarity of the calibration coefficients can be made by the EOFs. The Principal Component Regression as we use it is not so much dependent on the individual EOFs. We rather use the PCs to fit our model and the EOFs are just used as a way to reduce the dimensionality of the predictors in only a few patterns. With the leading EOFs we can reconstruct the original anomaly pattern and therefore using 10 EOFs will lead to >90% of variance represented, irrespective to the period the EOFs are based on (i.e. $X\_i = \text{sum } pc\_i,j * eof\_j$) ] Regarding the investigation of the stationarity of the teleconnections the reviewer is right. Therefore, we will perform the required calculations and discuss these results in the text.

3) In the conclusions and the abstract it is mentioned that the marine network can produce skillful spatial reconstructions for the eastern NA basin. But even in this area there seems to be a massive underestimation of the amplitude. I think this underestimation in general could be described more in both the text and in the abstract. 4) I guess the reconstructions are best in the areas close to the proxies. But I don't think this is discussed much in the paper. The position of the proxies could be indicated on the maps. 6) There has been a discussion in the literature of the reconstruction methods ability to get the amplitude right. Methods with temperature as the dependent variable, as those used here, are prone to underestimate the variability (Christiansen and Ljungqvist 2017, doi: 10.1002/2016RG000521 and references therein). The results in this paper seem to get an underestimation of the variability even in the case with noise-free proxies. The reason for the underestimation of the variability should be investigated and discussed.

Regarding the major comments 3, 4 and 6 we plan to elaborate more on these in the discussion, abstract and conclusions as the reviewer suggested. The positions of the proxies are plotted in all of the maps shown, but as we now acknowledge it is hard to see. Therefore we plan to improve the figures of the main text.

5) It would be nice to see a plot of the time-series of the 5 real-world proxies. This would

also allow the reader to judge if the AR1 process used for the pseudo-proxies is sound. By the way, I am surprised that the authors did not show a real-world reconstruction based on the 5 Arctica islandica.

The 5 Arctica time-series are already published and we have cited those papers for the interested readers in the lines 97-100. However, showing these series here would not help to decide whether the AR1 process to represent the noise in sound or not. For that purpose, an analysis of the residuals resulting from a regression between the real proxies and the local water temperatures would be required. Those series are short, and thus it would be in any case difficult to decide if an AR1 model is enough. However, in the revised version we will additionally perform the experiments using additional noise realizations of the AR1 model to better account for the spread in the noise within the different proxy locations.

Minor comments: line 35: I think the abbreviations such as NA and PPE should also be defined in the text and not just in the abstract. line 54: Perhaps another word than "aggravating" should be used here. line 92: grid point -> gridded. line 111 and 30: It is confusing that "reconstructions" are used for different things. line 119: Why 1999 and not 2005? line 142: A degree sign is missing. Section 2.2.1: It would be better if single-letter symbols would be used in the formulas instead of Proxy, EOF, CC etc. line 193: The sentence beginning with "The key .." does not make much sense to me. l205: The pseudo-proxy review Smerdon 2001 (doi: 10.1002/wcc.149) should be cited somewhere. line 200: With only five (or less in section 3.3) proxies it does not seem to make any sense to make a EOF transformation of the proxies and keep all five modes. This step is usually done to reduce the number of degrees of freedom which is not necessary here. This should be discussed. line 223: What are the correlation of the other 4 proxies? It could be noted that values

We agree with the reviewer regarding all of the minor comments. Concerning the comment on line 200, Canonical Correlation Analysis identifies the pairs of co-variability patterns between the North Atlantic SSTs and the proxy SSTs that have maximum

temporal correlation. CCA involves the inversion of the co-variance matrix of each field and although it can theoretically be applied to the original fields, co-variance matrices of geophysical fields tend to be near-singular and therefore its direct inversion leads to numerical instabilities. Some sort of regularization is needed and this is usually achieved by a prior EOF analysis. The reviewer is right that for the proxy field with only 5 records it does not make a difference, but the EOF analysis of the proxies just simplifies the inversion of the co-variance matrix, since it brings it to a diagonal form.

The comment related to the single-letter symbols is in our opinion a matter of taste, but we agree with the reviewer to have a consistent naming of the symbols throughout the text. Therefore we will address these issues in the revised version of the manuscript.

---

## Author Comment (AC4) · 21 Jun 2017

Answer by M. Pyrina et al. to: Anonymous Referee #3

We would like to thank Referee #3 for the constructive comments. We will try to be clearer in the introduction about the broader significance of the work and expand our analysis including more validation statistics as the reviewer suggested. Also, we would like to thank the reviewer for the technical corrections.

Specific comments:

1) Why choose calibration periods that are earlier than the validation period? Given

that every real-world climate reconstruction would be calibrated during the later period during which instrumental data is available, wouldn't it make more sense to calibrate the PPEs using the 1850-1999 period and then validate on the earlier time periods (Medieval, LIA, etc)? To me, that seems more intuitive and would still address the stationarity issue.

We agree with the reviewer that every real-world reconstruction would be calibrated in recent times in order to reconstruct past times, but we conducted the experiment in this way in order to be able to additionally compare our model-based pseudoproxy results with the pseudoproxy results based on the reanalysis data. Furthermore, this approach gives us the opportunity to perform tests on calibration periods that are considered to be different in their climatic background state. Although from a practical point of view the choice of the real-world calibration period is constrained by the availability of meteorological observations, the climate models allow us to circumvent these issues and carry out sensitivity experiments in the context of the virtual world of the climate model.

2) Is there any reason why these two particular CFR methods (PCR and CCA) were chosen, while other common CFR methods (e.g., RegEM-TTLS and RegEM-ridge) were excluded? I'm not necessarily suggesting that the authors need to redo the analyses with additional CFR methods, but I would at least like to see a little justification for why these methods were chosen while others were excluded.

We agree with the reviewer and therefore plan to further justify the selection of the two methods in the introduction. These methods are widely used in paleoclimate reconstructions. Therefore, we used these two methods in order to check how sensitive the results are depending on the analysis we choose.

3) I think it would be useful to include other complementary validation statistics, such as mean bias, coefficient of efficiency (CE), reduction of error (RE), and/or root mean squared error (RMSE). I'm not sure that only correlation and standard deviation ratio

are enough for a robust assessment of model performance. A spatial assessment of mean bias and either CE or RMSE could add important information to this study.

We agree with the reviewer that complementary validation statistics could add important information to this study and therefore we plan to include the Reduction of Error (RE) and the Root Mean Squared Error (RMSE).

4) How is it possible to have a standard deviation ratio greater than 1 for the CCA results? As I understand it, all parametric reconstruction methods will result in at least some variance loss unless the proxy is perfectly correlated with the target climate variable (McCarroll et al. 2015), which would only be the case at the particular grid cells with the noise-free pseudoproxies. Even in other studies that used noise-free pseudoproxies (e.g., Smerdon et al. 2010, 2011), at least some variance loss was observed. I therefore don't see how the reconstructed SSTs could have grid cells with greater variance than the "observed" (or in this case, model-simulated) SSTs.

Thanks to the reviewer we identified a small error in the scripting that does not change the conclusions of this manuscript, but due to that the SD ratio regarding only the CCA results was slightly overestimated in the regions lacking proxies. In the revised version the necessary corrections will be done.

5) The authors state that they chose to retain the first 10 PCs in PCR and the first 5 EOFs in CCA. How sensitive are the results to this choice? How were these thresholds chosen? Why were different thresholds chosen for the two CFR methods? As it stands, these seem like arbitrary choices. Did the authors consider more objective criteria for determining these thresholds, such as the "estimated noise continuum" approach used by Mann et al. 2007 or an optimization approach similar to Smerdon et al. 2010?

For the PCR method we retained 10 EOFs because they represent more than 90% of the spatial co-variance of the North Atlantic SSTs and in this way we capture most of the NA SST covariance. In the CCA method we retained 5 EOFs, as 5 is the maximum number of EOFs that we can keep in the case of the proxy field, because this number

depends on the number of proxy locations. We will add some lines of explanation in the method section.

6) I would like to see some more discussion about the stationarity of the PPEs. Specifically, there is an implicit assumption when creating the pseudoproxies that the proxy response to SST variability is stationary (since the pseudoproxies are just SST+noise). Is this necessarily a realistic assumption for real-world A. islandica, or is it possible that the response of this species to SST variation could be non-stationary (similar to the well-known "divergence problem" in high latitude tree-ring widths)?

It might indeed happen that the response of a living animal changes through time, but so far there is no known "divergence problem" in Arctica islandica. However, there may be other potential sources causing non-stationarity in the response of bio-physiological proxies (turbidity, salinity, food availability) or basic changes in the ecosystem functioning which are not accounted for in our approach testing for stationarity. These questions are difficult to model statistically, as the non-stationarity may arise with very different character and has not been clearly characterized in real proxies, as the dendroclimatological divergence problem illustrates. We will briefly discuss how a pseudo-proxy test could address some non-stationarities, e.g. by introducing AR1 coefficients that are themselves random variables.

---

## Author Response (AR1)

Dear Editor, dear Referees,

Thank you for evaluating this manuscript. Your comments have led to several improvements of this work. In the revised version of the manuscript, additionally to the suggestions of the reviewers, we have included further analysis and discussions of the results according to the previously provided "author's first response" to the reviewers. The results of the additional analysis are shown in the Appendix, from Figure 10S to Figure 22S.

In the following, you can find a point by point reply to the Referees' comments regarding the specific changes included in the revised version of the manuscript. The Referees' comments are shown in bold letters, while our replies are denoted by italic fonts. The point by point reply to the Referees' comments is followed by a marked-up manuscript version that includes all relevant changes highlighted in the text with grey colour.

**Point-by-Point actions**

**Referee #1**

**Major comment:**

**The most important comment I have from a substantive standpoint relates to the definition of the Z noise variable in Equation 6. This nature of this random variable is critical to how the noise degradation for the pseudo-proxy experiments is defined. It needs to be described for the readers, so they can make judgements for themselves about the experimental results.**

*-The Z noise variable is described now in more detail in lines 256-260: "White noise series, $Z_t$, were generated in order to take into account the randomness of the noise. In a second step the white noise $Z_t$ series were transformed into red noise using the AR(1) process described in Eq. (6). The AR(1) noise series were finally re-scaled to the corresponding amplitude to achieve the desired relative variance of noise in the pseudoproxy record (see e.g. Smerdon, 2012). The pseudo proxy record is then composed of the sum of the simulated grid-cell temperature anomalies record and the AR1 noise series."*

**Specific Comments:**

**Line 25: There are many more studies that can be cited here, including those by Mann et al. (2008, 2009), Hegerl et al. (2007), etc. A more full sampling of these should be included here. The same is true for the global and regional spatial reconstructions in line 26.**

*-A more in depth literature review can be found in the revised version of the manuscript. (Specifically see lines: 25-28, 38, 63, 68, 77, 81).*

**Line 37: Similar to the comment on the prior page, are there other studies that could be cited here?**

*-The work of Wang et al., (2017) has been added (line38): "Terrestrial proxy records were also used by Wang et al., (2017) for the reconstruction of a 1200-year AMO index."*

**Line 41: Italics are often used to denote scientific names for living creatures.**

*-The name: "Arctica islandica" is now denoted with italics throughout the text.*

**Line 43: The word "archive" is generally used to indicate the kind of geological deposition situation or the kind of living being (such as a tree) from which a climate proxy can be extracted. In this sense, A. islandica is not an archive, but is rather a proxy extracted FROM an archive.**

*-Here we disagree with the reviewer. Arctica islandica is a living being. It is a physiological entity, in the same sense as a tree, therefore an archive according to the definition suggested by the reviewer. The proxy derived from Arctica islandica is related to the width of the growing increments or to the isotopic composition within the individual growth bands that Arctica islandica forms.*

**Line 45: The standard nomenclature for denoting a species abbreviation is to abbreviate the genus name to its first letter, which is capitalized.**

*-Instead of using an abbreviation, we will refer to the name of the mollusk constantly as "Arctica islandica".*

**Line 50: Again, additional citations of existing work should be added here. Mann et al. and Luterbacher et al. work in particular are worth mentioning as key thrusts in this regard.**

*-The work of Mann et al. and Luterbacher et al. has been added in the first lines of the introduction (lines 25-28).*

**Line 75: It would be good to cite some of the Smerdon et al. work on PPEs here.**

*- The work of Smerdon et al. 2012 is added (line 79), while studies of Smerdon et al. have been generally added in the introduction (i.e. Smerdon et al. 2010, 2013 in line 68)*

**Line 106: Define what "bucket correction" means.**

*- Line 119-122 has been changed to:* "Data up to 1941 were bias-adjusted using "bucket correction", as the SST measurements were performed using a variety of buckets including canvas buckets and better insulated wooden or rubber buckets (Hirahara et al. 2014). The mean temperature change experienced by the water collected in a bucket until the temperature measurement is performed can be estimated and the temperature measurement can then be bias-corrected."

**Line 133: Should references like these have the years included in ( ) ?**
*- The reviewer is correct. The reference format has been changed accordingly throughout the manuscript.*

**Line 168: Are both the original and reconstructed data detrended, or just the reconstructions? It would seem strange if it is just the reconstructions, and this would need to be explained.**
*- Both the original and reconstructed data are detrended. Therefore the word "detrended" has been added in the line now being the 186.*

L**ine 172: Is the EOF itself the eigenvector in this nomenclature?**
*- We choose the name "EOF" to represent the eigenvectors. To be clearer in that matter, the sentence in line 196 has been changed to: "Each eigenvector is represented by a spatial pattern (EOF, Empirical Orthogonal Function) which is associated with a temporal evolution (PC, Principal Component)."*

**Line 178: The coefficient being referred to here is NOT the estimated one (which is properly mentioned in Equation 3), but rather the actual one that is TO BE ESTIMATED, and so the "hat" should not be used here.**
*- The reviewer is correct. Therefore, the "hat" has been removed.*

**Line 179: Equation (2) gives the underlying model, and thus the "hat" over the coefficient should not be employed here.**
**Line 186: Equation (3) is the ESTIMATED form of Equation (2), and thus a "hat" over the coefficient should be employed here.**
**Line 203: Please double-check that the corrections made in line 203 are accurate. I believe these corrections are accurate, as the predictand here should be the CC^NA variables, NOT the CC^pr variables.**
*- The reviewer is correct in all the cases above (lines 204, 211, 229) and changes have been made accordingly.*

**Line 218: Add a reference here for the Y-W equations.**

*- A reference has been added (line 247; von Storch and Zwiers, 2001).*

**Line 225: Would it be useful to evaluate an AR(2) process also as a suitable candidate to mimic the behavior of relevant noise for A. islandica? I don't know this for a fact, but think it is worth considering and at least mentioning in this context. More generally, one could potentially not need to make a specific decision about the lag nature of the AR process. Alternatively, one could utilize something like Hosking's algorithm, which inputs the entire spectrum of the Autocorrelation Function (ACF) to generate a simulated random time series with the same ACF structure as the original time series. A useful resource in this regard is the "hosking.sim" function in R. I'm not saying that evaluations in this way is a requirement, but offer it as something to think about, as its generality could be useful in this context.**

*-The AR1 model is the simplest model to describe the autocorrelation of non-climatic processes within the proxy record, such as the accumulation of nutrients that are used in the next growing season. Other mechanisms may rise longer memory autocorrelations, such as interaction with other species within the ecosystem that may give rise to non-climatic oscillations in the growth rate. However, these have barely been studied even in the dendroclimatological context which is much more mature than in the realm of marine proxies. A test with more complex noise structures would seem justified when empirical studies of the behaviour of Artica islandica offer further hints about those processes.*

**Line 225: KEY**
**What value of Z noise was added, and how was this estimated? This component of the process model in Equation (6) is not further explained, and needs to be known for interpretation of the noise actually added to the perfect pseudo-proxies.**
*-The Zt noise series are now described with more detail in lines 256-260.*

**Line 277: Briefly mention why the anti-correlation may have no physical meaning.**
*-The lines 314-316 have been changed to: "These results indicate that the CCA method would be problematic regarding the reconstruction of the NA SSTs over those areas, based on the r2 and r3 realizations of the MPI-ESM model, as an anti-correlation between the original and the reconstructed temperature evolution has no physical meaning."*

**Line 332: Again, are there more recent sources that can be cited here. These are 15+ years old, and additional and more advanced model simulations are available. The reviewer is not an expert in this regard, but would like to ask if there are no studies of NAO teleconnection variability based on the more recent model experiments.**
*-More recent studies have been added (Line 384).*

**Line 335: Explain why poor representation would lead to generally low reconstruction skill in the NA basin when utilizing covariance-based CFRs.**
*-The lines 385-388 have been changed to: "However, as CFRs are covariance-based approaches, a poor representation of large scale climatic patterns that describe spatial relationships between different regions of the NA basin would lead to a generally low reconstruction skill in the basin, which is not consistent with the results of this study."*

**Line 356: If possible given the nature of the two models, attempt to provide a numerical ratio to characterize the difference in magnitude of the spatial resolutions.**
*-We have added an approximation in kilometres regarding the models resolution in section 2.1.2 (see lines 139 and 158).*

**Line 359: Since volcanic eruptions are not regular occurrences, is this difference between the models really a salient factor here?**
*-This is not a salient factor for the differences between the models, but it could be a potential factor influencing the calibration interval.*

**Line 363: Provide figure references for the reader to see these results.**
*-Figure references have been added in line 415.*

**Line 381: Loss of variance is also a fundamental characteristic of least squares regression itself, unless there is an exact linear relationship between the predictand and predictor variables, with the error term in the regression model equal to zero.**
*-The reviewer is correct, therefore line 435 has been reformulated to: "Moreover, the truncation of the eigenvalue spectra by the PCA and CCA methods could lead to variance losses (Smerdon et al., 2010)."*

**Line 650: The sites of the pseudo proxies should be indicated in a least one of these panels, but likely better would be to show them in all the panels, for all similar graphics. On a much more blown-up look, I see that there are three circles near Britain, one near Iceland, and one at far-northern Norway. If these are the sites, that should be made evident in the caption or in the figure panels somewhere. These circles are hard to see, even blown-up to a large degree, and so they should be filled in rather than just an outline, perhaps in green or black color.**
*-The reviewer is correct. Therefore, the sites are now clearly seen in all the figures included in the main text. The circles are not filled, because we want to show in every case the values of correlation and of the other metrics on the specific location of the individual sites.*

*-Also, we would like to thank Referee #1 for the technical corrections made throughout the manuscript.*

**Referee #2**

*-Regarding the general comments of Referee #2 about the broader significance of this work, we have added some lines in the Introduction and the Conclusions are changed:*

*Lines 92-94: "The potential of reconstructing the marine climate without the usage of terrestrial proxies is tested here for the first time using a network of annually resolved marine proxies over the NA basin. The performance of commonly applied CFR reconstruction methods is tested in different surrogate climates suitable for PPEs."*

*Lines 438-450: "The tested CFR methods can have an impact on the spatial skill of the reconstruction, especially when noise contaminated pseudo-proxies are used. Therefore, it is important to assess CFR methods with noise-contaminated proxies rather than with ideal noise-free ones. Moreover, regardless of the model simulation used, both CFRs methods are affected by loss of reconstruction variance. Field correlations are high on the east Atlantic basin where less variance loss is observed and a better prediction skill is indicated by the RMSE and RE. The CCA method is problematic when a significantly low number of proxies is used (two and three proxies), but the spatial skill of the reconstruction using CCA and five proxy locations is similar to the results calculated with the PCR method. Even though the models used as the basis of the PPEs were previously evaluated in the context of CFRs, it is found here that the most important differences pertaining the spatial skill of the reconstruction are caused by the choice of model simulation used , rather than by the specific CFR method or by the calibration period. With our PPEs we have demonstrated that the SSTs can be skilfully reconstructed not only around the proxy sites of Arctica islandica but in a broader area of the eastern Atlantic basin. This is an important result in the context of spatially resolved CFRs in the NA basin that lacks annually resolved marine proxy data, as a small sized proxy network of the marine bivalve mollusk Arctica islandica (five proxy locations) can provide valuable information for SST reconstructions of the northeast Atlantic Ocean."*

**Major comments:**

**1) The pseudo-proxy experiments seem to be done for only one realization of the AR1 noise even though several models are used. Many of the conclusions – which are based on rather small differences in the correlation maps – could be different if another noise realization was used as shown in, e.g., Christiansen et al. 2009 (doi:10.1175/2008JCLI2301.1). Preferably an ensemble of realizations should be used. Alternatively different realizations should be performed and the (possible) differences discussed.**

*-An additional AR(1) realization was used and the results are commented on lines 321-326: "The results of this section were also calculated with the PCR and CCA methods, but for the contamination of the pseudo-proxies an additional realization of the AR(1) model (Eq. 6) was used. The results are given in the Appendix (Fig. 14S, Fig. 15S, Fig. 16S and Fig. 17S). This analysis confirms the conclusions drawn by the CFRs using the initial AR(1) noise realization, for all reconstruction skill metrics, as well for the importance of the model simulation in comparison to the calibration period and the reconstruction method."*

**2) Both the considered methods depend on EOF analysis and section 4.1 discusses the stationarity of the calibration coefficients. However, the paper does not include any comparison of the EOFs or the spatial correlation structure. I would advise the authors to compare the EOFs from the different data-sets and the different periods. The stationarity of the teleconnections could also be investigated by looking at the map of correlations between a grid-point with proxy-data and all other points.**

*-We have investigated the stationarity of the teleconnections. The results are given in lines 368-372: "At inter-annual time scales, and for each calibration period used in this study, we calculated the teleconnections of the NA SSTs to the regions co-located to the Arctica islandica sites, using the COBE2 reanalysis data and the output of the MPI-ESM and CCSM4 model. Generally, models and reanalysis data show that the regions that exhibit high and statistically significant correlations (r≥+0.8) exist mainly between the proxy sites and their surrounding waters and that these teleconnections are stable in time (see in Appendix; Fig. 18S-Fig22S)."*

**3) In the conclusions and the abstract it is mentioned that the marine network can produce skillful spatial reconstructions for the eastern NA basin. But even in this area there seems to be a massive underestimation of the amplitude. I think this underestimation in general could be described more in both the text and in the abstract.**

*-Regarding that matter we have added the line 20: "Both methods are appropriate for the reconstruction of the temporal evolution of the NA SSTs, even though they lead to a great*

*loss of variance away from the proxy sites" in the abstract and the lines 440-442 in the conclusions: "Moreover, regardless of the model simulation used, both CFRs methods are afflicted by loss of reconstruction variance. Field correlations are high on the east Atlantic basin where less variance loss is observed and a better prediction skill is indicated by the RMSE and RE."*

**4) I guess the reconstructions are best in the areas close to the proxies. But I don't think this is discussed much in the paper. The position of the proxies could be indicated on the maps.**
*-The position of the proxies is now indicated on the maps. Regarding the reconstruction skill close to the proxy locations see lines 446-448: "With our PPEs we have demonstrated that the SSTs can be skilfully reconstructed not only around the proxy sites of Arctica islandica but in a broader area of the eastern Atlantic basin."*

**6) There has been a discussion in the literature of the reconstruction methods ability to get the amplitude right. Methods with temperature as the dependent variable, as those used here, are prone to underestimate the variability (Christiansen and Ljungqvist 2017, doi: 10.1002/2016RG000521 and references therein). The results in this paper seem to get an underestimation of the variability even in the case with noise-free proxies. The reason for the underestimation of the variability should be investigated and discussed.**
*-Regarding the underestimation of variability see lines 431-436: "The decrease in variance can be expected for CFR methods that are based on linear regression and combine the signal of the targeted variable with unrelated variability (Smerdon et al., 2011; von Storch et al., 2004). In the case of ideal pseudo-proxies a proportion of variance is lost due to the disturbance term (Eq. 2) that is not modelled during the reconstruction (Eq. 3), while in the case of noise-contaminated pseudoproxies additional variance loss is expected due to the non-perfect correlation between the pseudoproxy and the local climate variable (McCarroll et al., 2015). Moreover, the truncation of the eigenvalue spectra by the PCR and CCA methods could lead to additional variance losses (Smerdon et al., 2010)."*

**5) It would be nice to see a plot of the time-series of the 5 real-world proxies. This would also allow the reader to judge if the AR1 process used for the pseudo-proxies is sound. By the way, I am surprised that the authors did not show a real-world reconstruction based on the 5 Arctica islandica.**
*- Regarding this comment, we have given a detailed answer to the previously provided "author's first response" to the reviewers.*

**Minor comments:**

*-We would like to thank Referee #2 for the minor comments and technical corrections. All technical corrections have been made, while the detailed responses to the minor comments can be found in the previously provided document "author's first response" to the reviewers.*

**Referee #3**

*-Regarding the general comments of Referee #3 about the objectives and key results of this work, we have added some lines in the Introduction (lines 92-94) and the Conclusions are changed (lines 438-450).*

*Lines 92-94: "The potential of reconstructing the marine climate without the usage of terrestrial proxies is tested here for the first time using a network of annually resolved marine proxies over the NA basin. The performance of commonly applied CFR reconstruction methods is tested in different surrogate climates suitable for PPEs."*

*Lines 438-450: "The tested CFR methods can have an impact on the spatial skill of the reconstruction, especially when noise contaminated pseudo-proxies are used. Therefore, it is important to assess CFR methods with noise-contaminated proxies rather than with ideal noise-free ones. Moreover, regardless of the model simulation used, both CFRs methods are affected by loss of reconstruction variance. Field correlations are high on the east Atlantic basin where less variance loss is observed and a better prediction skill is indicated by the RMSE and RE. The CCA method is problematic when a significantly low number of proxies is used (two and three proxies), but the spatial skill of the reconstruction using CCA and five proxy locations is similar to the results calculated with the PCR method. Even though the models used as the basis of the PPEs were previously evaluated in the context of CFRs, it is found here that the most important differences pertaining the spatial skill of the reconstruction are caused by the choice of model simulation used , rather than by the specific CFR method or by the calibration period. With our PPEs we have demonstrated that the SSTs can be skilfully reconstructed not only around the proxy sites of Arctica islandica but in a broader area of the eastern Atlantic basin. This is an important result in the context of spatially resolved CFRs in the NA basin that lacks annually resolved marine proxy data, as a small sized proxy network of the marine bivalve mollusk Arctica islandica*

*(five proxy locations) can provide valuable information for SST reconstructions of the northeast Atlantic Ocean."*

**1) Why choose calibration periods that are earlier than the validation period? Given that every real-world climate reconstruction would be calibrated during the later period during which instrumental data is available, wouldn't it make more sense to calibrate the PPEs using the 1850-1999 period and then validate on the earlier time periods (Medieval, LIA, etc)? To me, that seems more intuitive and would still address the stationarity issue.**

*-Regarding this comment, the lines 178-182 were added: "Although a real world reconstruction would be calibrated during an overlapping period between the proxy data and the instrumental data, in this study earlier calibration periods were also tested in order to be able to compare the model-based pseudoproxy results with the pseudoproxy results based on the reanalysis data. Furthermore, this approach gives us the opportunity to perform tests on calibration periods that are considered to be different in their climatic background state and judge the effect of these periods on the reconstruction skill."*

**2) Is there any reason why these two particular CFR methods (PCR and CCA) were chosen, while other common CFR methods (e.g., RegEM-TTLS and RegEM-ridge) were excluded? I'm not necessarily suggesting that the authors need to redo the analyses with additional CFR methods, but I would at least like to see a little justification for why these methods were chosen while others were excluded.**

*-A justification regarding the methods can be found in lines 99-101. Some new lines of justification have been as well added (lines 101-104): "Other linear methods widely used in the context of spatial CFRs include the RegEM algorithm of Schneider 2001 and the Analog Method, but none of those methods has been found to outperform the CFR techniques chosen to be tested in the current study (Christiansen et al., 2009; Gómez-Navarro et al., 2017; Smerdon et al., 2016; Smerdon et al., 2008)."*

**3) I think it would be useful to include other complementary validation statistics, such as mean bias, coefficient of efficiency (CE), reduction of error (RE), and/or root mean squared error (RMSE). I'm not sure that only correlation and standard deviation ratio are enough for a robust assessment of model performance. A spatial assessment of mean bias and either CE or RMSE could add important information to this study.**

*-Complementary validation statistics were performed. The results can be seen in the Appendix Figures 10S-17S and in the text in the lines 292-296: "The RMSE of the*

*reconstructed SST evolution obtains generally low values close to the proxy sites for both methods and for different calibration periods, indicating that in those regions the prediction capacity is better (Appendix, Fig. 10S and Fig. 11S). The RE patterns (Appendix, Fig. 12S and Fig. 13S) follow similar spatial patterns to the ones shown by the correlation maps (Appendix, Fig. 3S and Fig. 2S), depicting higher values over the eastern Atlantic basin especially for the PCR method."*

*Also in lines 319-320: "Areas where field correlations are high, exhibit the highest values of RE (~0.5, see Appendix; Fig. 12S and Fig. 13S) and generally low values of RMSE (Appendix, Fig. 10 and Fig. 11)."*

**4) How is it possible to have a standard deviation ratio greater than 1 for the CCA results? As I understand it, all parametric reconstruction methods will result in at least some variance loss unless the proxy is perfectly correlated with the target climate variable (McCarroll et al. 2015), which would only be the case at the particular grid cells with the noise-free pseudoproxies. Even in other studies that used noise-free pseudoproxies (e.g., Smerdon et al. 2010, 2011), at least some variance loss was observed. I therefore don't see how the reconstructed SSTs could have grid cells with greater variance than the "observed" (or in this case, model-simulated) SSTs.**

*-Thanks to the reviewer a small error in the scripting was identified and the results regarding the SD ratio for the CCA method were corrected.*

**5) The authors state that they chose to retain the first 10 PCs in PCR and the first 5 EOFs in CCA. How sensitive are the results to this choice? How were these thresholds chosen? Why were different thresholds chosen for the two CFR methods? As it stands, these seem like arbitrary choices. Did the authors consider more objective criteria for determining these thresholds, such as the "estimated noise continuum" approach used by Mann et al. 2007 or an optimization approach similar to Smerdon et al. 2010?**

*-Lines 198-199 has been changed to: "In our analysis we kept the first 10 eigenvectors, as they represent more than 90% of variability and therefore most of the NA SST covariance is captured."*
*-Line 222 has been added: "The number of retained EOFs is equal to the maximum number of EOFs that can be kept in the case of the proxy field, as this number cannot exceed the number of proxy locations."*

**6) I would like to see some more discussion about the stationarity of the PPEs. Specifically, there is an implicit assumption when creating the pseudoproxies that the proxy response to SST variability is stationary (since the pseudoproxies are just SST+noise). Is this necessarily a realistic assumption for real-world A. islandica, or is it possible that the response of this species**

**to SST variation could be non-stationary (similar to the well-known "divergence problem" in high latitude tree-ring widths)?**

*- A more detailed answer to the reviewer has been previously provided in the "author's first response". The Lines 235-238 have been added in the text: "The response of Arctica islandica to the SSTs, as of other bio-physiological proxies, might be non-stationary due to basic changes in ecosystem functioning or due to changes in factors such as food availability, salinity and turbidity. These processes are difficult to model statistically, as the non-stationarity may arise with very different character and has not been clearly characterized in real proxies."*

**Technical corrections:**
*-We would like to thank Referee #3 for the technical corrections. All of them were taken into account.*

[revised manuscript text omitted]

---

## Author Response (AR2)

Dear Editor, dear Referees,

First, we would like to thank all reviewers for re-evaluating the revised version of our manuscript and the comments on the new version that helped to further improve the paper. Please find below a point-by-point response on the inquiries and suggestions on the revised version of the manuscript. The Referees' comments are shown in bold letters, while our replies are denoted by italic fonts. The point by point reply to the Referees' comments is followed by a marked-up manuscript version that includes all relevant changes highlighted in the text with grey colour.

**Point-by-Point actions**

**Referee #1**
REPORT#1

**Specific comments:**

**-Abstract: Without conducting tests on real-world A. islandica proxies, I'm not sure that the final sentence of the abstract is entirely justified based on the results presented in this manuscript. Since this study is conducted entirely in the virtual reality of a pseudoproxy experiment (with no testing of real-world A. islandica), I don't think it is justified to say that "The results show that the marine network of Arctica islandica can be used to skillfully reconstruct the spatial patterns of SSTs at the eastern NA basin." This statement assumes that real-world A. islandica proxies behave like the pseudoproxies. Based only on the results presented in this study, this seems like a big assumption to make, particularly since the construction of the pseudoproxies assumes both stationarity and linearity, whereas the real-world proxies may not have a stationary or linear relationship to SST. Further, if the real proxies behave like the noise-contaminated pseudoproxies used here, then the skill of SST reconstructions may actually be quite limited (as suggested by relatively low RE values in Figs. 12S and 13S and quite high variance losses, even in locations relatively close to the proxy locations). The results of this study show that the choice of surrogate reality makes a big difference and that both CCA and PCR could be appropriate choices for reconstructing SST in this region, but I think the authors should be a little more circumspect (both in the abstract and in the conclusion) about what exactly this study tells us about the skill of a real-world reconstruction of SST with A. islandica.**
*-The last sentence of the Abstract ("The results show that the marine network of Arctica islandica can be used to skillfully reconstruct the spatial patterns of SSTs at the eastern NA basin") refers to the marine network of Arctica islandica, not to the real world proxy itself. The study showed that the choice of the surrogate reality makes a difference, but in any case the pseudoproxies "sampled" at the locations where Arctica islandica is collected,*

*provide with valuable information about the eastern Atlantic basin (Correl. coef. ~ 0.6 for the noise contaminated experiment). We agree with the reviewer that the variance loss is due to the non-perfect correlation between the pseudoproxy and the local climate variable, but it is also due to the linear regression methods used for the reconstruction and that is already shown by the ideal-pseudoproxy experiment. Following the reviewers comment we have revised this sentence both in the abstract and in the conclusion.*

*Lines 21-23: Under reasonable assumptions about the characteristics of the non-climate noise in the proxy records, our results show that the marine network of Arctica islandica can be used to skillfully reconstruct the spatial patterns of SSTs at the eastern NA basin.*

*Line 449-451: This is an important result in the context of spatially resolved CFRs in the NA basin, which lacks annually resolved marine proxy data, as a small sized proxy network of the marine bivalve mollusk Arctica islandica (five proxy locations) could provide valuable information for SST reconstructions of the northeast Atlantic Ocean.*

**-Line 83: "CFR" is misspelled "CRF."**
**-Line 187: should read "Reduction _of_ Error"**
**-Line 236: "…non-climatic noise, _it_ is usually…"**
*-We have applied the changes in the lines 83, 187 and 236 in the manuscript.*

**-Are equations 2 and 3 correct? I could be wrong, but I think to reconstruct PC_{i,t}, you would sum across all j, not across all i. So I think Eqns. 2 and 3 should have SIGMA_{j=1-n} (where n is the number of proxies) rather than SIGMA_{i=1-10}. Again, I could be wrong, but the authors should double-check these two equations and confirm that they are indeed correct.**
*-The reviewer is correct and the equations have been changed.*

**-I would highly recommend a couple minor modifications to the figures that will make them more easily readable and interpretable. First, the figure labels are so small that they are nearly impossible to read. The font should be larger. Second, the color scales on all of the SDR figures should be rescaled. Theoretically, SDR should never be greater than 1, so it isn't necessary to have a color scale that goes up to 2. That just makes it harder to distinguish the spatial variation of SDR. Also, since SDR ranges from 0-1, a sequential color scheme would probably be most appropriate, with light shades representing low SDR and dark shades representing high SDR (as opposed to the currently-used diverging color scheme, which implies a distinct split in the data that are being mapped).**
*-We agree with the reviewer. We will re-plot the SDR main text graphs with a more appropriate color bar. Moreover, the figure labels are not needed as the individual figures are framed within a table. Therefore, we will remove the individual figure labels.*

**-RE figures: it's unclear from the caption and colorbar what the black grid cells represent. Are these regions with RE<0? The spatial patterns of the black grid cells also look a little weird. There are some subplots with entire regions blacked-out, and some where the black grid cells instead form lines or unfilled polygons. I'm also not sure that the colorbar should stop at a minimum of zero, since RE can be much lower than zero. It looks like**

there are probably regions of the NA (particularly far from the proxy locations) where RE is in fact less zero, but it is hard to tell based on the current scaling of the colorbar. I think it should be shown much more clearly where RE is less than zero.
*-We agree with the reviewer and we will replot the figures 12S and 13S.*

**-Figure 1: Correlation coefficients are "columns 1 and 3," not "columns 1 and 4," right?**
**-Figures 1 and 2: it should be clearer in the captions that these are for noise-free pseudoproxies.**
**-Figure 7S: Is this caption supposed to say "As in Fig. 6S" rather than "As in Fig. 5S"?**
*-The reviewer is correct and the comments regarding the Figures 1, 2 and 7S have been taken into account.*

**Referee #2**
REPORT#2

**The authors have considered my comments satisfactorily and I am now ready to suggest that the paper is accepted. I have a few additional minor comments that I will urge the author to consider but my acceptance of the paper does not depend on it.**

**18: The sentence "The addition of noise .. " is a little strange. The noise in the proxies is important for the realism of the experiments. So perhaps something like "Larger differences in skills appear when the proxies are noisy".**
*-We agree with the reviewer, therefore the sentence has been changed to: "Conducting PPEs using noise contaminated pseudo-proxies instead of noise free pseudo-proxies is important for the evaluation of the methods, as more spatial differences in the reconstruction skill are revealed."*

**25: Christiansen and Ljungqvist 2016, gives an overview over methods and results regarding hemispheric and global reconstructions.**
*-We think that this sentence has already a lot of studies cited. Moreover, the study of Christiansen and Ljungqvist 2016, has been cited in other parts of the manuscript.*

**55: More precisely, the estimate of the uncertainty can be underestimated if not the screening process is included in this estimation.**
*-We are not sure we understand the point of the reviewer. It seems to be the same to what Line 55 states: "The magnitude of the reconstruction uncertainties can vary depending on the different time scales (Briffa et al., 2001) and can be possibly underestimated if the proxy records have been previously screened according to their high covariance with instrumental records."*

**104: Is "co-located" the right word? The point is that the proxy sites should be "inside" the gridded fields?**
*-We believe that the word co-located is the correct one, because it represents the grid box that surrounds the real world proxy location.*

**204: This is the problem with choosing the proxy as the independent variable. There is no physical interpretation of the noise.**
*-Following this comment, we made changes in the line 204: "The error could be an unobserved random variable that adds noise to the linear relationship between the dependent variable (PC) and the regressors (Proxy SSTs), and includes all effects on the regressors not related to the dependent variable (Christiansen, 2011)."*

**lines 427-432: I don't think this is precise. The problem is basically that proxies are used as independent variables as described in Christiansen and Ljungqvist 2016.**

[revised manuscript text omitted]